# Photoinduced bond oscillations in ironpentacarbonyl give delayed synchronous bursts of carbonmonoxide release

Ambar Banerjee [1✉], Michael R. Coates [1], Markus Kowalewski[1], Hampus Wikmark [2], Raphael M. Jay [2], Philippe Wernet [2] & Michael Odelius [1✉]

Early excited state dynamics in the photodissociation of transition metal carbonyls determines the chemical nature of short-lived catalytically active reaction intermediates. However, time-resolved experiments have not yet revealed mechanistic details in the sub-picosecond regime. Hence, in this study the photoexcitation of ironpentacarbonyl $Fe(CO)_5$ is simulated with semi-classical excited state molecular dynamics. We find that the bright metal-to-ligand charge-transfer (MLCT) transition induces synchronous Fe-C oscillations in the trigonal bipyramidal complex leading to periodically reoccurring release of predominantly axial CO. Metaphorically the photoactivated $Fe(CO)_5$ acts as a CO geyser, as a result of dynamics in the potential energy landscape of the axial Fe-C distances and non-adiabatic transitions between manifolds of bound MLCT and dissociative metal-centered (MC) excited states. The predominant release of axial CO ligands and delayed release of equatorial CO ligands are explained in a unified mechanism based on the $\sigma^*$(Fe-C) anti-bonding character of the receiving orbital in the dissociative MC states.

[1] Department of Physics, Stockholm University, AlbaNova University Center, SE-106 91 Stockholm, Sweden. [2] Department of Physics and Astronomy, Uppsala University, Box 516, SE-751 20 Uppsala, Sweden. ✉email: ambarpchem@gmail.com; odelius@fysik.su.se

Transition metal carbonyls have a rich legacy of well-known photochemistry[1–3]. The nature of the metal carbonyl bond and its cleavage under photoexcitation is a fundamental problem in chemistry with applications in catalysis and synthesis[1]. The photochemistry of more complicated cases of transition metal carbonyls containing metal-metal bonds opens alternative reaction pathways of energy relaxation[4]. There has been a renewed interest in these compounds with the advent of ultrafast spectroscopic techniques, which provide means to precisely measure the initial photophysics and ultrafast evolution of the electronic structure[5,6]. The choice of especially first-row transition metal carbonyls for these studies is motivated by their relative simplicity in terms of electronic structure and chemistry, due to the small number of electrons and limited relativistic effects. Of these systems, ironpentacarbonyl $Fe(CO)_5$ has been one of the most thoroughly investigated systems[7], in particular the processes initiated by photoinduced metal-to-ligand charge-transfer (MLCT). This includes attempts to study the ultrafast molecular dynamics (MD) using UV pump–probe and ionization techniques[2,8]. Trushin and coworkers established, in their femtosecond dynamics study, the sequential dissociation of carbon-monoxide (CO) from $Fe(CO)_5$ into $Fe(CO)_4$ and subsequently $Fe(CO)_3$, with the first step happening within 100 fs[8]. The generation of $Fe(CO)_4$ from $Fe(CO)_5$ has also been established using ultrafast electron diffraction studies, and there the gas phase dynamics is suggested to follow a singlet pathway leading to the generation of $Fe(CO)_4$ in an electronic singlet state[9]. Time-resolved valence and core-level photoelectron spectroscopy employing an X-ray free-electron laser has been used to firmly establish the sequential and singlet dissociation pathway of $Fe(CO)_5$ in the gas phase[10]. Ramasesha and co-workers have also very recently confirmed this singlet dissociation pathway using ultrafast IR spectroscopic techniques, and have estimated that triplet involvement and inter-system crossing happen in 15 ps time frame[11]. From these studies, insights into the kinetics of the process and the chemical changes based on charge localization were also obtained[12]. The initial ultrafast dynamics of these systems determine how the photon energy is channeled into electronic and nuclear degrees of freedom, and eventually leads to the preferential formation of one reactive intermediate over another. The events initiating these processes are still not at all well understood primarily due to limited temporal resolution in the experiments. Hence, explicit simulations of the excited state molecular dynamics (ESMD) can yield key understanding of vibrational excitations and non-adiabatic transitions following the MLCT excitation and reaction pathways leading to the photodissociation.

Theoretical studies have been used to gain understanding of systems similar to $Fe(CO)_5$, but these also lack a detailed description of early time scale dynamics[13]. The excited states of $Fe(CO)_5$ have been investigated with advanced quantum chemical methods[14,15]. With similar accuracy, the pathways of the ground state $Fe(CO)_5 \rightarrow Fe(CO)_4 + CO$ reaction have been studied by Roos and co-workers[16]. Insight into the sub-picosecond dynamics in photochemistry has direct implications for the understanding of the photochemical reactivity of these complexes, many of which have broader applications[17]. The photochemical C–H activation of transition metal carbonyls is a prime example of this[18]. The photodissociation of CO ligands from metal carbonyls, especially in iron complexes of which $Fe(CO)_5$ is a prototypical case, is of interest to a broader audience of chemists as it has direct implications on the binding of CO molecules to biologically important iron complexes in solution[19]. In addition, $Fe(CO)_5$ has been extensively studied lately, not only for photolytic processes, but for interesting ground state phenomena[20,21]. These studies include the direct observation of the transition state for Berry pseudorotation[22]. The photochemistry of $Fe(CO)_5$ in the solvent phase is of great interest and has been studied using femtosecond X-ray (resonant X-ray inelastic scattering—RIXS) spectroscopy[23,24] indicating competing pathways of inter-system crossing and $Fe(CO)_4$-solvent complex formation. With the advent of ab initio ESMD techniques[25] these exact problems and dark areas, which could not be addressed in previous experimental studies due to limited temporal resolution, can now be investigated theoretically. This will in turn provide a challenge for the experimental improvement of temporal resolution and the development of new experimental techniques altogether[25–28]. Over the past few years, a wide range of problems dealing with complex photochemistry/photophysics of transition metal chemistry has been investigated by MD techniques which include photosensitizers and spin relaxation dynamics[29,30]. Analogous to the $Fe(CO)_5$ system, $Cr(CO)_6$ has been studied with excited state MD simulations using forces from time-dependent density functional theory (TDDFT) to follow dynamics in the lowest singlet excited states $S_1$, $S_2$, and $S_3$[31].

In this study, we access the early stages of photochemistry/photophysics of $Fe(CO)_5$ using ESMD[25] with a semi-classical treatment of non-adiabatic transitions. The $Fe(CO)_5$ complex has a trigonal bipyramidal ($D_{3h}$) ground state geometry with distinct axial and equatorial CO ligands, subject to slow exchange from Berry pseudorotation[22]. After an MLCT excitation, we observe several unique features and mechanistic intricacies in the $Fe(CO)_5 \rightarrow Fe(CO)_4 + CO$ photodissociation process. This includes an unprecedented report of synchronous bursts of CO at periodic intervals of ~90 fs determined by the potential energy shape of bound MLCT states. The simulations shed light on the mechanistic pathway involving photodissociation predominantly of axial CO and vibration relaxation involving several nuclear degrees of freedom. We also find a unique correlation between the dissociation time and angular distortion towards $C_{4v}$ symmetry in the $Fe(CO)_5$ unit which indicates an alternate mechanistic pathway. Both the preferential loss of axial CO and the delayed dissociation at the distorted geometries, of axial and equatorial CO, can be understood from a frontier molecular orbital analysis. Population analysis reveals that the Fe–C bond dissociation involves non-adiabatic transitions from MLCT states to metal-centered (MC) states associated with an internal electron redistribution leading to repulsive forces.

## Results

### UV–vis spectrum and which electronic states are initially populated. Previous studies[3,8] suggest that the experimentally employed 267 nm irradiation induces a $10e' \rightarrow 4e''^*$ orbital transition and targets a MLCT state ($^1A_2''$). Low lying electronic states and frontier orbitals of $Fe(CO)_5$ are shown in Fig. 1 (extended summary in Supplementary Fig. 1), following the nomenclature of the $D_{3h}$ point group in accordance with the previous calculations[14]. For evaluation of our computational TDDFT framework (see "Methods"), we display experimental, reproduced from ref. [32], and calculated UV–vis spectra for gas-phase $Fe(CO)_5$ in Fig. 1a, along with the energies of various excited states computed at different levels of theory in Supplementary Table 1 and in Supplementary Figs. 2, 3.

The theoretically computed UV–vis spectra in Fig. 1a were obtained for the equilibrium geometry and with a Wigner phase space sampling of 300 geometries, and reproduce the presence of a peak in the spectral region corresponding to the MLCT $^1A_2''$ state. Apart from the consistent underestimation of all excitation energies, we notice in Supplementary Table 1 a qualitative agreement between TDDFT and high level quantum chemistry, differing essentially only in the ordering of the quasi-degenerate

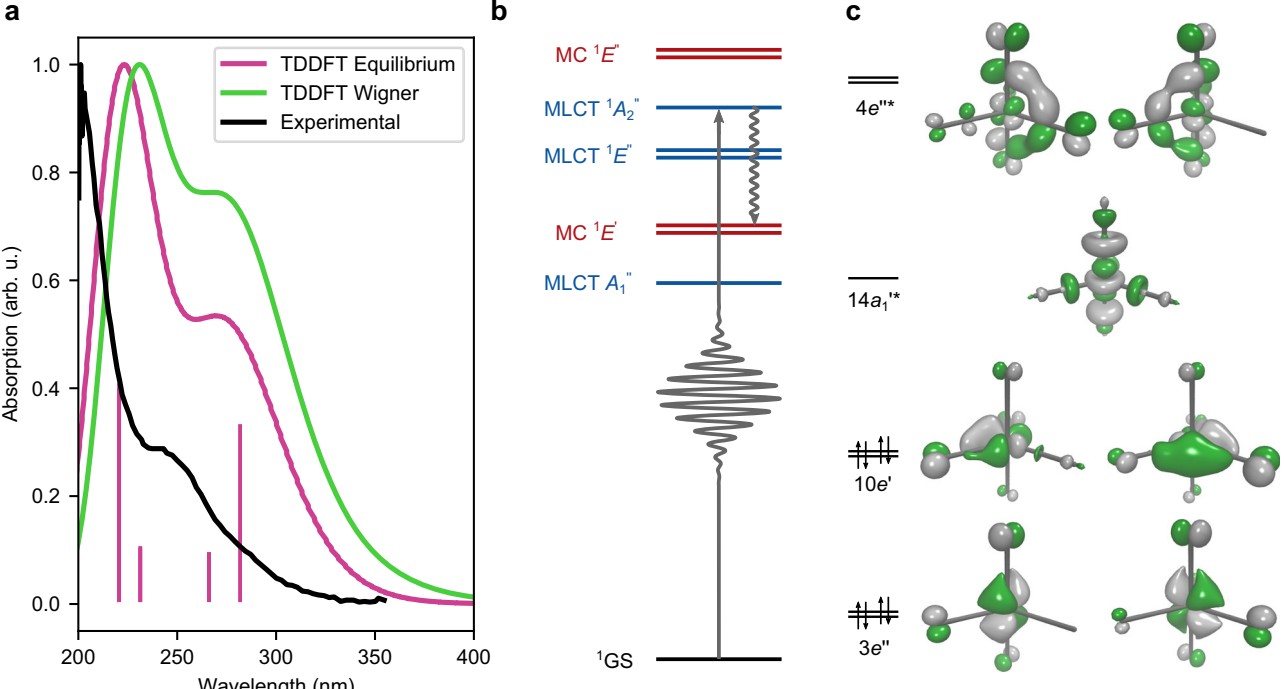

**Fig. 1 Photoexcitation, energy levels and frontier molecular orbitals of ironpentacarbonyl Fe(CO)$_5$. a** The time-dependent density functional theory (TDDFT) simulated UV–vis spectra for the equilibrium geometry (pink curve) and the Wigner distribution (green curve), and the experimental gas-phase UV–vis spectrum reproduced from ref. [32] (black curve). The pink sticks correspond to the discrete TDDFT transitions at the equilibrium geometry. **b** Level diagram of electronic states involved in the photoexcitation and in the subsequent dissociation process. The character of metal-to-ligand charge-transfer (MLCT) and metal-centered (MC) states is denoted by red and blue color, respectively. **c** Frontier molecular orbitals involved in the photodissociation of ironpentacarbonyl, with specifications of orbital symmetries and ground-state occupation.

bright $^1A_2''$ and dark $^1E'$ states, due to the different treatment of dynamical correlation, which earlier has been identified as being possibly involved in the excitation at 267 nm[3]. We also investigated the excitation with different functionals and found the results to be in close resemblance with CAM-B3LYP, see Supplementary Table 1. TDDFT and NEVPT2 give qualitatively equivalent UV–vis spectra as shown in Supplementary Fig. 2, with TDDFT underestimating the excitation energies. NEVPT2 and CASPT2 (with IPEA 0.25) give comparable energies and has been shown to overestimate the excitation energies, which is corrected at the CASPT2 with no IPEA(0.0) level of theory[33] and become closer to the TDDFT results. The energy difference between the bright $^1A_2''$ (MLCT) state and the dark $^1E'$ (MC) states are consistent for TDDFT and NEVPT2. With the excited state dynamics being mostly confined to those states, and other MLCT and MC states having similar shapes, we expect the excited state dynamics observed in the TDDFT framework to be physically relevant.

A multiconfigurational approach in the ESMD simulations would be preferable, but as seen in Supplementary Table 1, the complete active space self-consistent field method (CASSCF) yields excitation energies that are too high due to the lack of dynamical correlation and n-electron valence state perturbation theory (NEVPT2) is too computationally demanding. Hence, we resort to TDDFT, which gives results in good agreement with earlier advanced studies[14,15]. In Supplementary Fig. 3, cuts in the potential energy surface (PES) of electronic states show similar shapes in TDDFT and NEVPT2, although there are differences in the relation between bound and dissociating states. The crossings between MLCT and MC states occur at different bond lengths for the PES from NEVPT2 and TDDFT, which could affect the time-scales of the dissociation process even if the mechanism remains

the same. Similarities between NEVPT2 and TDDFT are seen for an example trajectory in Supplementary Fig. 4, consistent with the trends in Supplementary Fig. 3. We also notice that the system remains a single reference far along the dissociation coordinate, as seen in Supplementary Fig. 5, which reaffirms the applicability of TDDFT.

Encouraged by this evaluation, we carried out ESMD using surface-hopping in SHARC[34] based on TDDFT computations, as described in Methods Section, which reaffirms our choice of method (see Supplementary Notes 1 and 2 for further details on the comparison of TDDFT/CAM-B3LYP and NEVPT2).

**Excited state molecular dynamics.** ESMD simulations in the singlet manifold were carried out on the 116 initial conditions from the Wigner sampling, that correspond to excitation of the $S_6$ state, involved in the bright MLCT transitions. Triplet states were excluded from the dynamics as the previous experiment based on ultrafast electron diffraction[9], photo-ionization mass spectrometry[8], XPS[10], and most recently time-resolved IR studies[11] have consistently ruled out ultrafast triplet pathways. However, it is noteworthy here that ultrafast inter-system crossing (ISC) has been observed in non-dissociative processes in other iron-complexes acting as photo-sensitizer[35,36]. A plausible explanation for the lack of ISC in our system is discussed in Supplementary Note 3, in relation to Supplementary Fig. 6 on the basis of shapes of potential energy surfaces and nuclear wave-function overlap.

Population dynamics, non-adiabatic transitions, and geometric coordinates were analyzed for the resulting trajectories in order to derive a detailed insight into the early events of the photo-dissociation. Out of the 116 trajectories, 110 trajectories showed single Fe–C bond dissociation as seen in Supplementary Fig. 7.

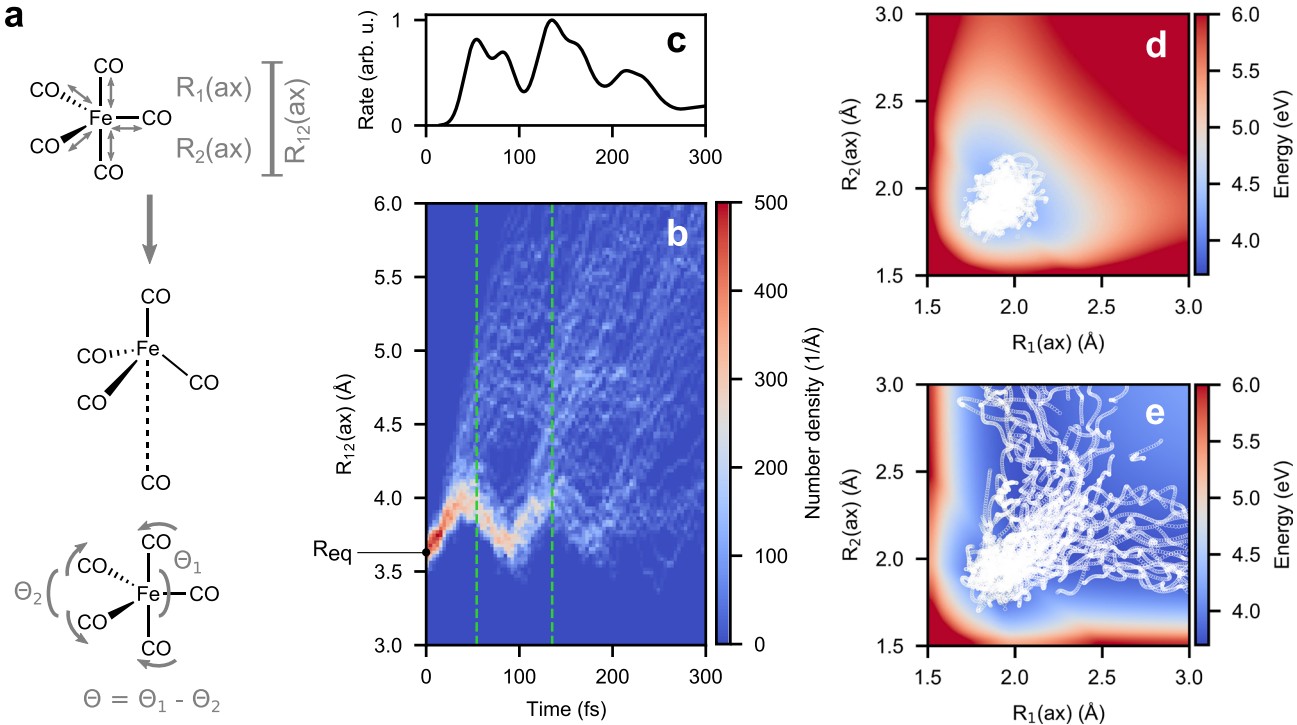

**Fig. 2 Release of axial CO from photoexcited Fe(CO)$_5$ synchronous with the symmetric stretch mode in axial Fe–C bonds. a** Schematic diagram for definition and relevance of the R$_{12}$(ax) coordinate in axial CO dissociation, and the angular difference ($\Theta = \Theta_1 - \Theta_2$). **b** Plot of distribution of R$_{12}$(ax) as a function of time for the 94 trajectories yielding axial CO release. **c** Rate of axial CO release as a function of time (obtained from the dissociation times $\tau_{dissoc}$) showing clearly visible periodic bursts ~90 fs apart. The rate curve is obtained by applying a smoothing Gaussian convolution of the distribution of $\tau_{dissoc}$ with a full-width half-maximum of 10 fs. **d, e** Two-dimensional cuts in the potential energy surfaces (PES) of S$_6$ (top) and S$_3$ (bottom) states obtained by time-dependent density functional theory (TDDFT) scans along the axial Fe–C R$_1$(ax) vs R$_2$(ax) distances. The white scatter overlaid on the PES shows segments populating these states in the trajectories in (**b**).

This statistically corroborates the experimental finding that indeed Fe(CO)$_5$ initially yields a Fe(CO)$_4$ fragment as suggested by Trushin and coworkers[8]. The double dissociation trajectories and trajectories which do not undergo any dissociation were excluded from further analysis due to insufficient statistics. Secondly, and very interestingly, we found that for events of single bond dissociation, 94 trajectories (85%) undergo release of an axial CO ligand and the remaining 16 trajectories exhibit release of an equatorial CO ligand (15%). From a careful inspection of all the individual trajectories, we introduce a criterion for dissociation, based on a running averaging over the Fe–C distance of the released CO group (see examples in Supplementary Fig. 8). The running average quenches fast oscillations in the Fe–C distance, which allows us to define a unique dissociation time ($\tau_{dissoc}$) for each trajectory, based on a threshold of 2.5 Å. The value of 2.5 Å is the lowest Fe–C distance being crossed only once in each of the 110 trajectories. This 2.5 Å criterion also makes sense when looking at PES in Supplementary Fig. 3, where the dissociative MC states are almost flat, and beyond that multi-reference effects take hold, as seen from Supplementary Fig. 5. The observations of bond oscillations and preferred axial CO release immediately raise a few questions. What is the mechanism for the major pathway of axial dissociation in terms of the states involved? Why is axial dissociation a preferred path? The fact that we have a minority pathway of equatorial dissociation, points to an alternate mechanism. How is that different from axial dissociation?

**Major channel—release of axial CO ligands.** Photoexcited Fe(CO)$_5$ exhibits multidimensional dynamics, but we highlight the most important coordinates to give a mechanistic picture of the

majority pathway. Analysis of the trajectories having axial CO release indicates that the dissociation process (see Fig. 2a) involves

i. a prominent elongation of the Fe–C bonds,
ii. followed by preferential elongation of an axial Fe–C bond, and
iii. angle distortion from $D_{3h}$ towards $C_{4v}$ symmetry

The initiating MLCT excitation induces vibrations in all the Fe–C bonds as seen in Supplementary Fig. 9. The preferential release of axial CO ligands, and the correlated vibrations in the axial Fe–C bonds (R$_i$(ax); $i = 1, 2$) motivated us to follow the motion in the reduced dimension of the R$_{12}$(ax) coordinate, defined as the distance between the two axial carbon atoms, as indicated in Fig. 2a. The distribution of R$_{12}$(ax) is plotted as a function of time after the excitation in Fig. 2b, in which we can distinguish the dissociating molecules from the motion in those remaining intact. We observe a prominent synchronous oscillation, corresponding to a breathing mode, with an amplitude of ~0.3 Å.

At regular intervals, there are bursts of dissociation of CO. The rate of dissociation as depicted in Fig. 2c, which we measure based on the dissociation criterion of the running average of the Fe–C distance crossing the 2.5 Å mark (see Supplementary Fig. 8), reveals three peak maxima around 50, 140, and 225 fs, which indicates that the release of CO is synchronous, with a period of ~90 fs. As discussed below in the context of equatorial dissociation, the angular distortion becomes active for later dissociation events.

**Energetics and electronic states.** Having characterized the major mechanistic pathway which predominantly leads to dissociation of

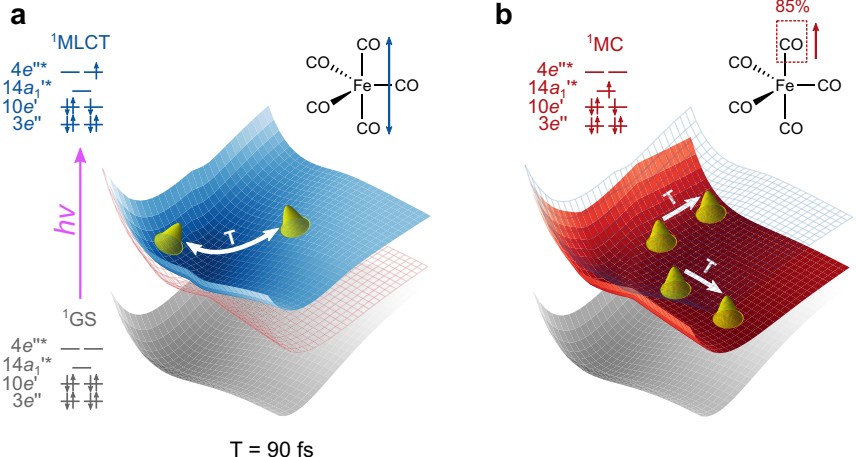

T = 90 fs

**Fig. 3 Schematic representation of the oscillation in the metal-to-ligand charge-transfer (MLCT) state and the periodic release of predominantly axial CO ligands in a metal-centered (MC) state. a** The oscillation of the two axial COs in the bound MLCT state occurs with a period of ~90 fs. **b** This leads to the periodic crossover to the dissociative MC states at the same frequency, which in turn produces periodic bursts of CO dissociation having a time lag of 90 fs. Orbital configurations of the ground state and specific MLCT and MC states are given.

axial Fe–C bonds, we proceed with analyzing energetics and electronic aspects; and to be more precise how do the energies of the different electronic states change along the reaction coordinate and in which state or group of states does the dissociation happen. For this purpose, we performed rigid two-dimensional (2D) PES scans of the $R_1$(ax) and $R_2$(ax) distances for the lowest ten electronic states. The adiabatic states can be partitioned into one class of bound excited states $S_5$ to $S_9$, represented by the PES of $S_6$ in Fig. 2d, and another class of dissociative excited states $S_1$ to $S_4$, represented by the PES of $S_3$ in Fig. 2e (for all states $S_2$ to $S_9$ see Supplementary Fig. 10). Analyzing the character at Fe–C > 2.2 Å of the adiabatic states $S_1$ to $S_9$, we notice that in the diabatic picture (see Supplementary Fig. 11), they are related to bound states with MLCT character, involving excitations into the $4e''^*$ orbitals in Fig. 1c, and four dissociative states with MC character, arising from the four $d-d$ excitations from the $3e''$ and $10e'$ orbitals into the $14a_1'^*$ orbital. To relate the molecular dynamics to the population dynamics, the 2D PES of each state was then overlayed with a scatter plot of the evolution in that state in the 94 axially dissociating trajectories. Figure 2d shows how the dynamics in $S_6$ (MLCT) is confined in the bound potential and tend to involve symmetric distortions, whereas in the $S_3$ (MC) state, which has a channel of dissociation along each $R_i$(ax) distance, the trajectories bifurcate into the release of either of the two axial CO ligands (see also Fig. 3). The observation of dissociation in the lowest electronic states corroborates the findings of earlier studies[3,8], which suggested that dissociation happens after non-radiative decay into dark MC states. The transition from the MLCT to a MC state is associated with a transfer of an electron from a ligand orbital ($4e''^*$) to a metal centered orbital ($14a_1'^*$), with a particular anti-bonding overlap symmetry between the Fe and the two axial CO groups, is what effectuates the dissociation process. This concept of the electron moving to an MO having a particular orbital overlap symmetry forms an essential part of the photo-dissociation dynamics.

In an attempt to generalize the insight from analysis of trajectories with the release of an axial CO ligand, we followed the energies of electronic states along the ESMD trajectories. Dynamically averaged potential energy cuts along the Fe–C coordinate, corresponding to the dissociating CO, are derived as the energies for each adiabatic state $S_i$ sampled as a function of dissociating Fe–C distance and averaged over the whole set of single dissociation trajectories and shown in Fig. 4a. The averaged potentials deviate from the rigid scan in Supplementary Fig. 3a,

since dispersion due to distortions along other degrees of freedom is also sampled. We also looked at the cases of 94 axial and 16 equatorial dissociation separately in Supplementary Fig. 12, and found no profound difference between them.

In this trajectory-based information, we clearly see two categories of excited states. Thus based on Fig. 4a, we confirm that $S_5$-$S_9$ are non-dissociative adiabatic states and $S_1$-$S_4$ are dissociative adiabatic states, in agreement with the analysis of the associated rigid scans shown in Fig. 2d, e and Supplementary Figs. 3 and 10. For dissociated geometries (Fe–C > 2.2 Å) $S_5$-$S_9$ have MLCT character and $S_1$-$S_4$ have MC character. As seen in a diabatic picture in Supplementary Fig. 11, the MLCT and MC states cross in the Franck-Condon region, which implies that the characters of the adiabatic states change.

The excitation puts the system in a manifold of non-dissociative states which results in correlated vibration of the two axial Fe–C bonds giving rise to a synchronous oscillation of $R_{12}$(ax) having a periodicity of roughly 90 fs. This synchronous oscillation in turn induces a periodic crossing of the point where non-adiabatic transition to the dissociative surface can happen. We can conceptualize this phenomenon as periodic leakage of the wavepacket from a dissociative to a non-dissociative potential happening with the same periodicity as that of the $R_{12}$(ax) oscillation in the non-dissociative surface, see Fig. 3. Thus this periodic leakage of the wavepacket from MLCT to MC gives the period bursts of axial CO release. We notice the similarity with non-adiabatical dynamics in neutral and ionic surfaces leading to periodically occurring dissociation in simple diatomics, like NaI[37], but to the best of our knowledge this phenomenon has not been previously reported for photodissociation in complex systems like metal carbonyls.

**Population dynamics and kinetic modeling.** To understand the underlying electronic mechanisms for the modes of dissociation and synchronous oscillation summarized in Figs. 2 and 3, we extracted the population dynamics from the simulations and performed kinetic modeling[38–40]. Since most of the trajectories exhibit early dissociation, the temporal analysis of electronic population in the different adiabatic states has been done from 0 to 300 fs. Notice that in about 20% cases the trajectory terminates before 300 fs following Fe–CO dissociation, due to the inability of the TDDFT framework to describe the Fe(CO)$_4$ species

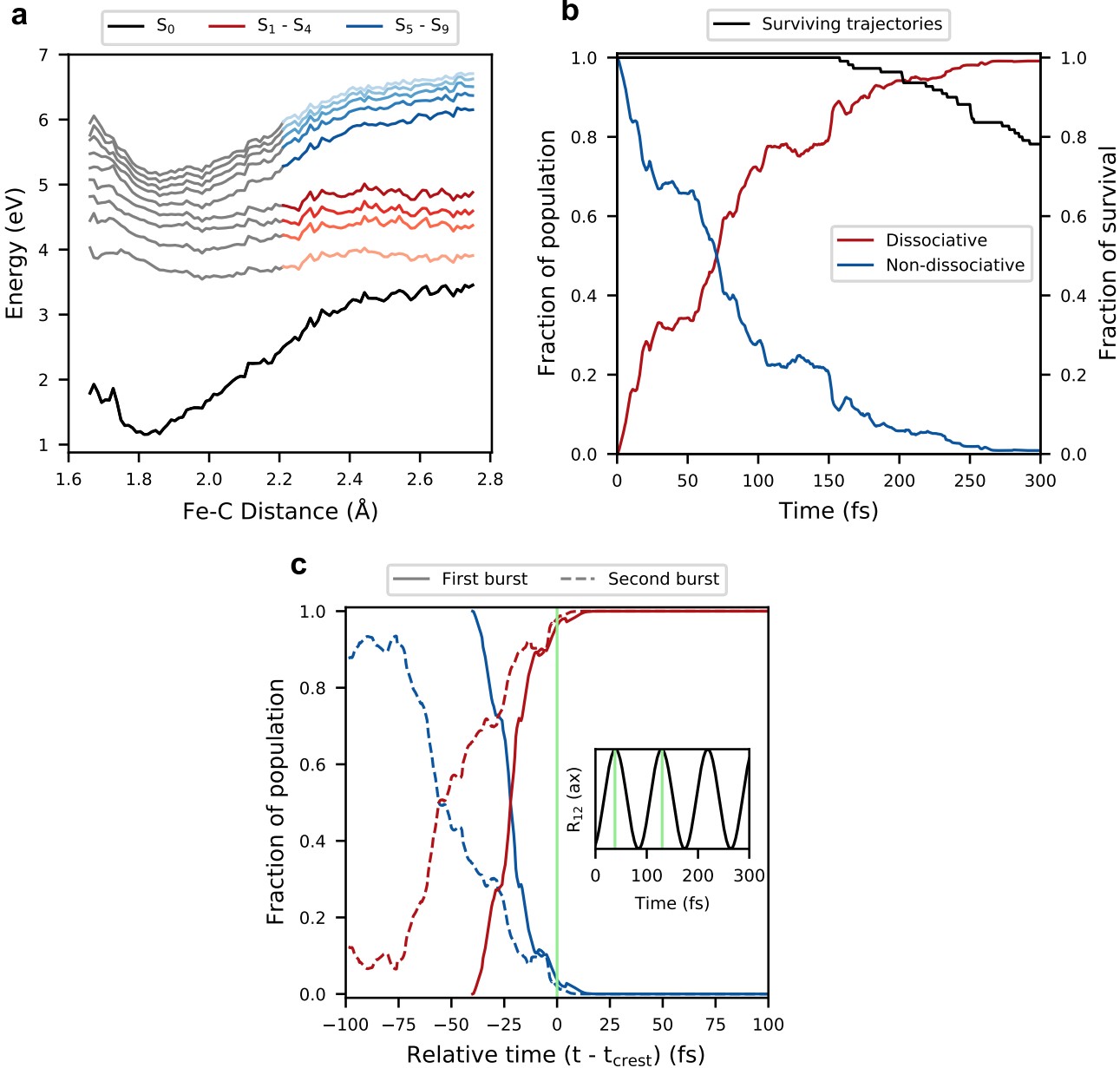

**Fig. 4 Electronic population dynamics for the photodissociation of Fe(CO)₅. a** The potential energy of all the states along the excited state molecular dynamics (ESMD) simulations were sampled as a function of Fe–C distance, corresponding to the released CO, and averaged over the 110 singly dissociative trajectories. At long Fe–C distance, non-dissociative metal-to-ligand charge-transfer (MLCT) and dissociative metal-centered (MC) states are colored blue and red, respectively. **b** Population dynamics of non-dissociative, ie. $S_i$ ($i = 5−9$), and dissociative, ie. $S_j$ ($i = 1$-4), states averaged over the 110 single dissociation trajectories. The black line represents the fraction of trajectories that survives as a function of time. **c** Normalized population dynamics for the two prominent bursts in subset of trajectories with axial dissociation from Fig. 2 presented on a relative time axis. The relative time 0 fs for first and second burst matches the corresponding crests of the oscillation of $R_{12}$(ax), respectively. The inset shows the schematic $R_{12}$(ax) oscillation and the green lines indicate the positions of $t_{crest}$ for first and second burst of axial CO ligands.

(see Fig. 4b). This is not a severe limitation, since the event of interest has then already been sampled. In Fig. 4b, we present an analysis of the population dynamics summed into bound and dissociative adiabatic states which gives a clear insight into the dissociation. The population of all the states as a function of time is presented in Supplementary Fig. 13. Since the adiabatic states, analyzed in Fig. 4b, are clearly separable into the MLCT and MC states for Fe–C > 2.2 Å, this corroborates the earlier experimental findings that photodissociation happens due to transfer from MLCT to MC states[3,8]. A state-specific analysis of population dynamics is discussed in the Supplementary Information on the basis state life-times in Supplementary Table 2 and the surface

hopping matrix in Supplementary Table 3. Non-adiabatic transitions occurring predominantly between adjacent adiabatic states i.e. from $S_n$ to $S_{n±1}$ motivates a kinetic model presented schematically in Supplementary Fig. 14 and discussed in Supplementary Notes 4 and 5, and shown to perform well in Supplementary Fig. 13 and in Supplementary Table 2.

To create a more firm link between the population analysis and the nuclear dynamics, we further analyzed the trajectories of axial CO release. Of these 94 trajectories with oscillations and bursts of the release of axial CO ligands seen in Fig. 2b, those dissociating in the first (27) and second burst (40) were analyzed separately. To understand how the non-adiabatic transition involving

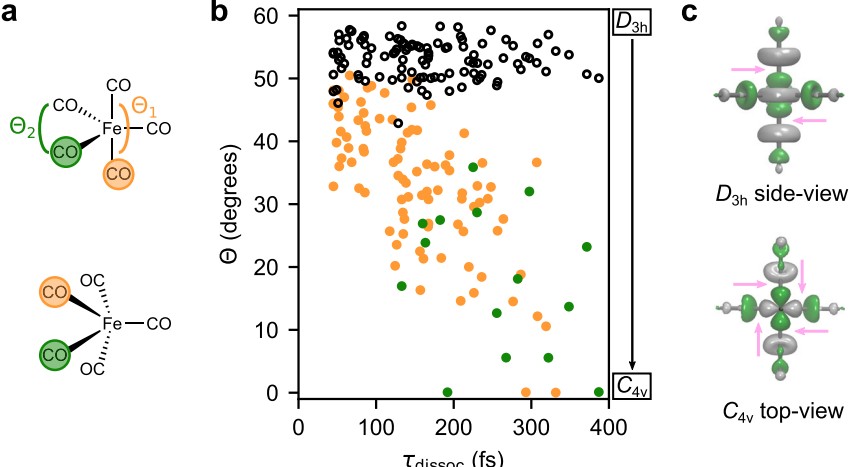

**Fig. 5 Correlation of angular distortion, from $D_{3h}$ geometry to near $C_{4v}$ geometry as measured by decrease in $\Theta$, and time taken for CO release.** **a** Schematic representation of degrees of freedom in the angular distortion parameter $\Theta = \Theta_1 - \Theta_2$, introduced in Fig. 2a, involving transition from $D_{3h}$ geometry to near $C_{4v}$ geometry; followed by release of CO. The green and orange spheres denote the axial and equatorial ligands which become equivalent upon angular distortion to the nearly $C_{4v}$ geometry. **b** Scatter plot of the smallest value of $\Theta$, before $\tau_{\mathrm{dissoc}}$, versus $\tau_{\mathrm{dissoc}}$ for all 110 singly dissociative trajectories. The orange and green solid dots represent release of axial and equatorial CO ligands, respectively. The hollow black dots show the initial distribution of $\Theta$ obtained from the Wigner distribution. **c** The excited singly occupied molecular orbital involved in the the dissociative MC states at ground state $D_{3h}$ geometry (top) and in the near $C_{4v}$ geometry (bottom) which is also the optimized geometry for the $S_5$ state. The pink arrows for both the cases indicate regions of anti-bonding overlap which effectuates the CO dissociation.

population transfer from the non-dissociative states to the dissociative states is related to nuclear motion, we introduce a time stamp at the crest of the $R_{12}(ax)$ oscillations. In Fig. 4c, we present the population dynamics for the first burst and the second burst of axial CO release, sampled relative to the time stamps. For trajectories which dissociate in the second burst, we see that there is a larger spread in time for the population transfer from the bound states to the dissociative states, in comparison to the population dynamics in the first burst. Because of the separation into a first and a second burst, we also see in the second burst an initial lag period before the increase in population transfer. This phenomenon of delayed population dynamics followed by sudden dissociation/cleavage of a bond, or ballistic dynamics, has been shown earlier for photolytic ring opening of dihydroazulene by Abedi and coworkers[41]. In their case, ballistic dynamics was proposed based on the existence of a similar profile of population dynamics involving a bound $S_2$ state, and a dissociative $S_1$ state. This indicates that the photodissociation of $Fe(CO)_5$ can be considered ballistic, since the population transfer from an initially excited MLCT to dissociative MC states is moderated by Fe–C bond length oscillations in bound states.

**Minority channels: equatorial dissociation and Berry pseudorotation.** As pointed out earlier, there is a 15% release of equatorial CO ligands. This warrants an investigation into the existence of a minority mechanistic route, parallel to the major axial dissociation mechanism discussed in Fig. 2. In the trajectories undergoing dissociation of an equatorial Fe–C bond, we observe that the initial dynamics starting from the $D_{3h}$ ground state geometry in the Franck–Condon region involves closing of the axial C–Fe–C angle and opening up of one of the equatorial C–Fe–C angles, similar to the notion of Berry pseudorotation[42].

The associated transformation from $D_{3h}$ to $C_{4v}$ symmetry can in general be quantified using the angular difference ($\Theta = \Theta_1 - \Theta_2$) introduced in Fig. 2a between the largest two C–Fe–C angles ($\Theta_1$ and $\Theta_2$ respectively) involving four unique Fe–C bonds in the complex, as depicted in Fig. 5a.

For the ground state $D_{3h}$ geometry the measure of $\Theta$ is 60° and the more $\Theta$ deviates from 60° the more the structure has moved towards $C_{4v}$ symmetry, which corresponds to the transition state geometry that we see in pseudorotation.

The pseudorotation leads to a loss of identity of equatorial and axial CO ligands. To get an idea of the degree of angular distortion that the system has undergone before dissociation, we measured the smallest value of the angular parameter $\Theta$ that is attained in each trajectory before the release of a CO moiety, i.e. reaching $\tau_{\mathrm{dissoc}}$. In Fig. 5b, we made a scatter plot of this smallest value of $\Theta$ with respect to $\tau_{\mathrm{dissoc}}$. To investigate the sensitivity to the initial geometry, we also plot the value of $\Theta$ in the Wigner sampling as a function of the dissociation time. We clearly see that the initial distribution for $\Theta$ has no effect on the time of dissociation, whereas the degree of maximum distortion, i.e. the minimum value $\Theta$ attains before dissociation, is roughly proportional to the dissociation time. In other words, those trajectories which dissociate later have time to undergo angular distortion. From the previous analysis of bound and dissociative adiabatic states, we can conclude that angular distortion modes are also activated and play an important role for the dynamics in the bound states. Hence, in the diabatic picture, the Fe–C bond oscillation and initiated pseudorotation occur in MLCT states.

At the ground state $D_{3h}$ geometry, the $S_5$ and $S_4$ states form a degenerate pair of $^1E'$ symmetry and MLCT character. However, this degeneracy is lifted along $e'$ vibrational modes, like the concerted closing (opening) of the axial (equatorial) C–Fe–C angles. Hence, this corresponds to a Jahn-Teller distortion mode. We optimized $Fe(CO)_5$ in the $S_5$ as a proof of principle and as the lowest representative for the bound MLCT states, since $S_5$ is bound across the PES, in contrast to $S_4$ which becomes a dissociative (MC) state as Fe–C distances elongates. At the optimized $S_5$ geometry, $Fe(CO)_5$ acquires a close to square pyramidal geometry with an axial C–Fe–C angle reduced to 165° and equatorial C–Fe–C angle opened to 140°. In Fig. 5c, we depict the frontier molecular orbitals, in which the excited electron resides in the dissociative MC states, for the ground state geometry ($D_{3h}$) and the $S_5$ optimized geometry (nearly $C_{4v}$

symmetry), because the $S_5$ state forms the lowest of the non-dissociative states.

We observed that the MO coefficients and the anti-bonding overlap are distributed over four CO moieties near $C_{4v}$ geometry, whereas it is localized on the two axial COs at the ground state ($D_{3h}$) geometry. The anti-bonding overlap, which is directly linked to the dissociation of the Fe–C bond, as discussed in detail in the next section, indicates predominantly axial dissociation for $D_{3h}$, but equal probability of axial and equatorial dissociation for $C_{4v}$. This geometry-dependent anti-bonding character of the metal-centered orbital ($14a_1'^*$ in $D_{3h}$ symmetry), receiving the excited electron in the dissociative MC states, explains the majority and minority mechanisms for photodissociation on a common basis. This unified mechanism of the photodissociation explains the time dependence in the relative release of axial and equatorial CO ligands seen in Fig. 5c.

## Discussion
From this theoretical study of the mechanism of the photodissociation of ironpentacarbonyl, we can reveal a mechanistic pathway closely related to conceptual ideas in the previous studies[2,8]. However, instead of non-adiabatic transitions between states with forces acting in different directions, as suggested in ref. [8], we observe oscillations of a Fe–C breathing mode in the MLCT state yielding regular bursts of CO release after non-adiabatic transitions to MC states. The initial fraction of dissociation is dominated by axial CO ligands, but due to excitation of a low-frequency pseudorotation mode equatorial and axial dissociation become equivalent. The Fe–C oscillations are associated with variations in the relative energies of the MLCT and ground states, which with sufficient time resolution could be detected experimentally[23,43].

Analysis of the rigid scans for both 1D and 2D PES, as in Supplementary Figs. 3, 10, and the population dynamics, shown in Fig. 4, along with visualization of the frontier molecular orbitals involved as the Fe–C bond dissociates, Supplementary Fig. 11 clearly indicates that the photodissociations happen by a transfer from non-dissociative to dissociative adiabatic states, which are associated with diabatic MLCT and MC states at long Fe–C distances. General chemists may find the orbital picture and arguments based on overlap symmetry of orbital more appealing than the state picture. Even in the case of angular distortion from $D_{3h}$ to near $C_{4v}$ geometries the same orbital overlap argument holds, which we have already discussed earlier. For trajectories which undergo pseudorotation-like distortion, the frontier molecular orbital accepting an electron in the dissociative MC states at the $S_5$ (MLCT) optimized geometry is shown in Fig. 5c lower panel. It is also interesting to see that the Fe–CO scan for both axial and equatorial CO from the $S_5$ (MLCT) optimized geometry is conceptually similar to the scan from $D_{3h}$ geometry (see Supplementary Fig. 15). We noticed from investigations of the vibrational modes in the $S_0$ (ground state) and $S_5$ (MLCT) geometries in Supplementary Table 4 that in the $S_5$ state there are symmetric axial/equatorial Fe–C stretching modes at ~400 cm$^{-1}$ corresponding to periods of ~80 fs, naturally associated the oscillation we observe, at both the $D_{3h}$ and $C_{4v}$ geometries. The vibrational signature of the CO moieties also shows sensitivity to the MLCT excitation, as shown in Supplementary Table 4, and could be used as an important experimental probe to investigate the dynamics in the MLCT states. We see that both the axial and equatorial CO photodissociation happens in the four lowest lying MC states.

On a conceptual level, we also want to mention analog to general reaction chemistry. The overlap arguments in Fig. 5c have a known parallel to ground state reactivity. The nuclear motion involved in the photo-induced release of axial CO, as shown in Fig. 2a, there is an uncanny similarity to that of a $S_N2$ reaction. The ground state geometry of Fe(CO)$_5$ has a $D_{3h}$ geometry which is similar to a $S_N2$ transition state. Similarly on the dissociative excited state surface for the axial Fe–C, this $D_{3h}$ geometry is a saddle point (see Supplementary Fig. 16), with the dissociative PES actually resembling half of the ground state PES of a classic $S_N2$ reaction. Thus, the Fe(CO)$_5$ molecule in a MC state at $D_{3h}$ geometry will follow the $S_N2$-like nuclear motion down hill on a dissociative PES. From an orbital overlap point of view, the $S_N2$ reaction happens as a consequence of populating a anti-bonding orbital as shown in Supplementary Fig. 16. In the MC state for Fe(CO)$_5$, the iron $3d_{z^2}$ orbital and the $\sigma^*$ orbitals of the CO groups are in anti-bonding combination, which are localized to axial COs for $D_{3h}$ geometry and delocalized on all four CO for near $C_{4v}$ geometries as discussed earlier, see Fig. 5c and Supplementary Fig. 17. This orbital overlap symmetry and population of anti-bonding orbitals (similar to the frontier molecular orbital situation in $S_N2$ reaction, see Supplementary Fig. 16), is what drives the release of axial CO from Fe(CO)$_5$ following similar nuclear dynamics as the $S_N2$ reaction.

To sum up our results are highlighted below in a point-wise manner:

i.   We found preferential axial CO release and only a minor fraction of equatorial CO release. This finding is a major step in the correct prediction of photodissociation of Fe(CO)$_5$ which can be addressed by different experimental techniques.

ii.  The population dynamics of states of different character has been described in detail, which can be directly accessed in spectroscopic studies.

iii. We have unearthed an unprecedented phenomenon, wherein CO dissociation happens in periodic bursts, as a consequence of periodically reoccurring transitions of the system from MLCT to MC states mediated by non-adiabatic coupling between them.

iv.  We have looked into the frontier molecular orbital character of the excited states, especially the dissociative states for Fe(CO)$_5$ and presented justification for the selective dissociation based upon anti-bonding orbital overlap symmetry. As schematically depicted in Fig. 3b the dissociation happens in reaching the MC states and the orbital involved in these states as shown in Fig. 5c clearly presents the picture how for $D_{3h}$ geometries the dissociation is preferentially axial and for near $C_{4v}$ geometries there is equal probability of dissociation of axial or equatorial Fe–C bonds.

Our ESMD simulation starts from the $S_6$ MLCT state and in principle the optical pulse used to photo-initiate the reaction is expected to also put the system in several other states depending on the band width. In Supplementary Table 1, we see that excitation into the $^1A_2''$ (MLCT) state will give the strongest transition. Hence, bright MLCT character could also be acquired by other close lying $S_7$ or $S_5$ states owing to the geometrical distortions in the Wigner sampling. These effects are neglected from the study and hence a direct comparison to the experimentally observed time constants are not warranted. This forms a limitation of our study on which further work is motivated, but looking at the diabatic potential energy surface in Supplementary 11 it is clear that all MLCT states have similar shapes and thus dynamics starting from on those surfaces can be expected to be similar to what we observe in the present case.

Hence, we conclude that both the majority channel of axial CO ligand release and the minority channel of delayed release of equatorial CO ligands can be understood in a unified mechanism of photodissociation, based on orbital anti-bonding character,

with conceptual similarities with the ground state $S_N2$ reaction. The points discussed just above have implications for future experiments which would i) verify whether axial or equatorial CO leaves during photodissociation, ii) gather spectroscopic signals of oscillation in the non-dissociative states and periodic photo-ejection, or in bursts, ejection of CO. The non-adiabatic couplings between bound MLCT states and dissociative MC states depend on their relative energies, and will hence vary with choices of transition metal and ligands. Hence, further simulations of different carbonyl complexes are required to establish general insights.

## Methods

**Ground state geometry and Wigner distribution.** Based on the CASPT2(12,12)/ TZVP optimized geometry, which is a gold standard, for $Fe(CO)_5$ as mentioned by Wernet et al.[23], vibrational modes were obtained at the B3LYP/cc-pVDZ// CASPT2(12,12)/TZVP level of theory using the MOLPRO package version Version 2012.1, interfaced to SHARC has been used[34]. Though the frequency calculation and optimization are done at different levels of theory no imaginary frequency was found, and performing frequency computation on the B3LYP/cc-pVDZ optimized geometry gives essentially the same vibrational spectra, differing only by ~20 cm$^{-1}$ in the five Fe–CO stretching modes, which play the most important role in the dynamics. We also checked the variation of geometry, at DFT level of theory and found that CAM-B3LYP, which is used for ESMD in this study, and TPSSH, which is often the best functional for transition metal system, predicts very similar geometry to the CASPT2(12,12) geometry. All these methods predicted the Fe–C(ax) = 1.80 Å and Fe–C(eq) = 1.81 Å. Density functional theory (DFT) is known to accurately predict the harmonic frequencies for single reference systems. Using the vibrational normal modes, a Wigner sampling of 300 points in phase space was created. The simulated UV–vis spectra were created from the discrete transitions in the TDDFT and NEVPT2 computations at the equilibrium geometry and from the 300 points of the Wigner sampling, followed by the application of Gaussian convolution of 0.8 eV full-width half-maximum.

**SHARC simulations.** For the ab initio ESMD using surface-hopping, SHARC version 2.1 has been used[25,34,44]. Using the Wigner distribution an absorption spectrum was calculated as formed by 7 singlet states. We avoided a larger number of excited states to enhance the fraction of trajectories initiated in $S_6$ state (determined by high transition dipole moment which corresponds to a MLCT transition), which primarily corresponds to the energy range of the commonly used experimental UV pulse. Inclusion of more higher lying states would invariably put a larger chunk of trajectories to the region at lower wavelengths and would not correspond to the experimental setup.

Further details on the ORCA calculations are discussed in the Methods (Quantum Chemistry) section. Of the 300 initial conditions in the Wigner sampling, we found that 116 trajectories could be excited into the $S_6$ state (having MLCT state character), following the protocol suggested in ref. [45] as implemented in SHARC[25,34]. These 116 trajectories were propagated from the $S_6$ state, now considering 10 singlet states. The choice of conducting the dynamics in only the singlet manifold is motivated from earlier experimental work which suggests a solely singlet pathway in gas phase[9–11]. Since the inclusion of triplet states would significantly increase the effort, we limited our investigation of the early stages of photodissociation in $Fe(CO)_5$ to include singlet states. The trajectories were run for up to 600 fs but many trajectories terminated much earlier at variable times due to convergence failure of the SCF or gradient module. However, each trajectory died a short period after the Fe–CO dissociation. The survival rate of the trajectories as a function of time is shown in Fig. 4b. All the analyses which are time dependent are done over these surviving trajectories only. Its noteworthy mentioning here that out of 110 trajectories only in 4 cases $Fe(CO)_5$ remained intact after 600 fs and hence the photodissociation of $Fe(CO)_5$ was captured properly. The non-adiabatic coupling between different states was handled by a local diabatization method using wavefunction overlap[46,47]. The simulation was performed using a time step of 0.5 fs. For electronic decoherence, the energy-difference-based correction[48] was used. The kinetic energy is adjusted by velocity rescaling. Standard SHARC hopping probabilities were used for surface hops. No scaling of energy or damping factors were employed. The rest of the parameter used by SHARC was taken to be default values.

**Quantum chemistry.** All computations on TDDFT and NEVPT2 level were done with ORCA 4.2.0[49], which also was used in an interface with SHARC[25,34] for the TDDFT based excited-state molecular dynamics simulations of the photo-dissociation of $Fe(CO)_5$. For the SCF in DFT, we used an energy convergence criterion of 1.0e$^{-8}$. For speeding up TDDFT computation during the excited state MD run, the Tamm-Dancoff approximation (TDA) was used. TDA was also used in the single point computation and scans, with single point computation of TDDFT without TDA on the equilibrium geometry produced essentially the same excitation energies. The potential energy cuts as shown in Supplementary Figs. 3, 6,

10, 11, and 15 are produced by unrelaxed (rigid) scan of one or two C–O bonds, respectively.

DFT has been shown to accurately reproduce the thermodynamics for CO dissociation for $Fe(CO)_5$ considering only the singlet manifold[50]. It is worth mentioning that TDDFT fails when multireference or near-degeneracy effects come into effect, but that occurs for the present system only after CO is released from the complex and the $Fe(CO)_4$ fragment is formed which has closely lying $S_0$ and $S_1$ states. Thus TDDFT can be safely used to study the dissociation process itself. Using TDDFT, UV–vis spectra and ESMD trajectories were computed with CAM-B3LYP/def2-TZVP functional[51,52] with RIJCOSX to make the single point computations faster.

For the CASSCF/NEVPT2/CASPT2/def2-TZVP calculations, we employed a (10e,10o) active space as employed by Pierloot and coworkers[15] and corroborates the findings of Daniel and coworkers[14]. The strongly contracted version of NEVPT2 as implemented in ORCA has been used. The fully internally contracted version of CASPT2 as implemented in ORCA with the imaginary shift of 0.2 and IPEA shift of 0.25 was used. In $C_1$ symmetry, state-averaged CASSCF calculations, including 7 singlets (for the UV–vis spectrum) and 10 singlets (for the singlet only potential energy surface and for the spot test along the TDDFT trajectories shown in Supplementary Fig. 4), followed by the perturbation theory computations. For producing the PES shown in Supplementary Fig. 6 which includes triplets we state-average over 10 singlets and 9 triplets without significantly affecting the singlet PES. The NEVPT2 spectrum is reproduced in Supplementary Fig. 2.

## Data availability

The data sets generated and analyzed during the current study are available from the corresponding author on request.

## Code availability

Codes generated during the current study are available from the corresponding author on request.

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

## Acknowledgements

M.O. acknowledges funding from the European Union's Horizon 2020 research and innovation program under the Marie Skłodowska-Curie grant agreement No. 860553. A.B. and M.O. acknowledge funding from the Carl Tryggers Foundation (contract CTS18:285), and M.K. acknowledges funding from the Swedish Research Council (grant agreement no. 2018-05346). The calculations were partially enabled by resources provided by the Swedish National Infrastructure for Computing (SNIC) partially funded by the Swedish Research Council through grant agreement no. 2018-05973.

## Author contributions

The project was designed and led by A.B. and M.O. in dialogue with Ph.W. and M.K. Simulations were performed and analyzed by A.B. and M.C. Kinetic modeling was performed by H.W. and R.J. The manuscript was written by A.B., M.C., and M.O. All authors contributed to the manuscript.

## Funding

## Competing interests

The authors declare no competing interests.
