## [Peer Review File · Nature Communications]

REVIEWER COMMENTS

Reviewer #1 (Remarks to the Author):

Banerjee et al. employed excited-state molecular dynamics to study the early dynamics of Fe(CO)₅ after excitation at 267 nm. The calculations were carried out using semi-classical simulations with SHARC at the TD-DFT level of theory. The calculations provide several detailed mechanistic insights into the early photophysics/photochemistry of Fe(CO)₅. Specifically, the calculations find, in support of earlier experiments, that a dark metal-centered state with anti-bonding Fe-CO character drives the dissociation of one CO from the ironpentacarbonyl complex. The calculations also predict excited-state driven oscillations in the axial CO release lifetime along with a rotation/pyramidalization mode that results in equatorial CO release.

Overall, the calculations appear to be carefully conducted and executed. My opinion is that this work is publishable. I have a few comments, mostly minor points:

1. Line 106: "Apart from the consistent underestimation of all excitation energies, we notice in Table S1 in the SI a close agreement between TDDFT and high level quantum chemistry, differing essentially only in the ordering of the quasi-degenerate bright 1A₂" and dark 1E' states, due to the different treatment of dynamical correlation, which earlier have been identified as being possibly involved in the excitation at 267 nm." → I don't know if I would call the TDDFT and NEVPT2 results a "close agreement." Not only is there a systematic redshift of TDDFT calculations by as much as ~1 eV, but the ordering of first 4 states also differ. Clearly, the NEVPT2 results seem to be in better agreement with the experiments, although the NEVPT2 calculation quality will likely quickly deteriorate for higher excited states due to the 10,10 active space used. However, the good news is that the energy difference between the bright 1A₂" (MLCT) and dark 1E' (MC) states are consistent for TD-DFT and NEVPT2, and those are the states largely driving the dynamics.

2. On a related note: please include computed oscillator strengths for each state's transition from the ground state in Table S1. It will make it easier to compare Table S1 and Fig S1.

3. Line 154: "the dissociation process (see Figure 2a) involves:

- i) a prominent elongation of the Fe-C bonds,
- ii) followed by preferential elongation of an axial Fe-C bonds, and
- iii) angle distortion from D_{3h} towards C_{4v} symmetry."

—> Despite how it is written, only point ii) can really be seen in Fig. 2a. Point i) isn't really clear in any of the figures (except in the SI) and point iii) is only clear later in Fig. 4.

4. On a related note, a few parts of the paper are difficult to navigate without referring to the SI.

For example try reading this without referring to the SI at each sentence: “Based on this we did kinetic modelling (see Figure S9 and discussion in the SI for details) for the population dynamics. The fitted curves, of electronic population are overlaid with the simulated data as shown in Figure S8. The area under fitted curves were taken as a measure of lifetime of different states (since due to the presence of back reactions exponential fits was not an option) and was found to match well with the lifetime data from the simulation (see Figure S2). This also establishes that the dynamics can be accurately described in a simple model, in which hopping from one state predominantly happens to the adjacent adiabatic states.”

I'm not sure how to best address this, since I appreciate that there are figure limit in the journal and that not all technical figures can be moved into the main text. Maybe, instead, such details should be moved to the Methods section at the end? Alternatively, maybe Scheme 1 could be moved to the SI and some of the more important figures from the SI could be moved into the manuscript. This is just a suggestion.

5. Instead of referring to the states as S1, S2, S3, etc., in several places in the manuscript it would be better to refer to the electronic character (MLCT, MC) or by symmetry labels where possible/appropriate. The reason is that the ordering of these states changes with geometry, computational method, etc. and so such labels are not really meaningful except to refer to the ordering of the states at the equilibrium geometry for a particular method. For example, near the equilibrium geometry S1-S4 are not dissociative, they only become dissociative once the CO metal bonds are elongated as can be seen in the diabatic surfaces in Fig. S7.

6. Related to the point above: the authors sample 300 initial conditions and only consider 116 which were excited into the “S6” state. If I understood this correctly, it means that the authors only chose states where S6 has 1A2” character (or a large oscillator strength) and excited into those, discarding the rest. In this case, they could be discarding states where other states, e.g., S7 or S5, have the 1A2” character due to some geometric distortions. In this way, the authors exclude trajectories that could potentially be interesting (where the molecule is excited along modes that reorder state energies) and bias the results of the dynamics.

I don't expect this will qualitatively affect the results of the calculations, but it could very well skew the relative yield of trajectories that give axial vs. equatorial dissociation or double dissociation.

7. I caught a few typos while reading the manuscript (likely, there may be a few more, just reporting those I noticed):

- line 26: there is no need for the comma after bonds.
- cam-B3LYP should be CAM-B3LYP (capital letters).
- line 246: "The fitted curves, of ..." → The comma is not needed.
- line 248: "exponential fits was" → "exponential fits were"
- line 379: anit-bonding → anti-bonding.
- line 409: "We also checked ... the CASPT2(12,12) geometry." → some commas are needed here, or split into two sentences.
- line 424: stats → states

Reviewer #2 (Remarks to the Author):

The present manuscript of Banerjee et al. presents simulations of the excited-state dynamics of the pentacarbonyl-iron complex in gas phase after UV excitation. Excited-state dynamics simulations of transition-metal complexes are still very challenging. Most past studies used quantum dynamics methods, that were, however, limited to only consider a few a priori selected degrees of freedom to describe the excited-state reactions. Only in the last few years, full-dimensional simulations using ab initio molecular dynamics methods such as used in this work have become possible for transition metal complexes, and the examples of these applications are still rare. Due to the small size and simple structure, pentacarbonyl-iron is a good test system for both experimental and theoretical methods. However, despite this, its excited-state dynamics involving CO dissociation after UV excitation are still not well understood. Thus, I think, the present study can be interesting to a wide audience of theoretical and experimental chemists and could be published in Nature Communications.

However, I have a concern regarding the role of triplet states in the dynamics, that should be clarified to validate the conclusions drawn in the study. In the present simulations, it was assumed that the CO dissociation takes place entirely in electronic singlet states. This assumption is based on earlier experimental studies (Ref. 9 and 10 in the manuscript) which suggest that the first CO dissociation occurs via a singlet pathway and that only later triplet states may become populated in the dissociated product. Previous simulations using ab initio MD or quantum dynamics have shown, however, that in iron complexes, intersystem crossing can occur on the (sub-) 100 fs time scale (e.g., JPCL 7, 2009, 2016; Inorg. Chem. 59, 14666, 2020; similar studies exist also for other 3d metal complexes). As this timescale is similar to that observed for the CO dissociation in the present

simulations, one may expect that intersystem crossing could also compete with CO dissociation in the singlet manifold in pentacarbonyl-iron. Thus, I think the authors should demonstrate that their assumption to include only singlet states is justified.

In addition, I have a series of minor concerns that should also be addressed (possibly also in the SI)

1. The documentation of the computational details needs to be more specific in several places:

- specify the version of the MOLPRO package

- details on parameters in DFT/TDDFT calculations: grid size, convergence parameters, etc., use of Tamm-Dancoff approximation (default in ORCA)?

- specify the default parameters in the SHARC simulations, e.g., decoherence correction, energy/momentum rescaling, etc... that are missing in the current computational details. Such defaults can change over time and should, thus, be documented in the studies where they are used.

- specify CASSCF/CASPT2/NEVPT2 variants: state-averaged CASSCF? Multi-state CASPT2? PC-NEVPT2? Did the authors use any shifts in the CASPT2 calculations? Also, a figure of the active orbitals would be welcome for the interested reader.

2. Figure 1: the relative intensity of the convoluted TDDFT equilibrium curve does not seem to match with the four shown transitions, where the second (higher-energy) absorption band has almost twice the intensity than the first one, while both bands seem to contain states with equal oscillator strength. How was this curve calculated?

3. p 4, l.111: the authors mention the close agreement between TDDFT results and high-level quantum chemistry results, although the excitation energies between CAM-B3LYP and NEVPT2 results differ by 0.7-1.4 eV in Table S1, which cannot really be considered good agreement anymore. Compared to the experimental absorption spectrum, it seems that CAM-B3LYP underestimates the excitation energies, while NEVPT2 overestimates the excitation energies. So, given both errors were systematic, this could explain the large differences and still justify the use of either method. However, I think the authors should try to demonstrate this. Are similar large differences between TDDFT and CAS methods for other 3d-transition metal carbonyl complexes known? NEVPT2 has been shown to give similar results as CASPT2 when using an IPEA shift of 0.25 au, and both tend to overestimate experimental excitation energies, at least for organic molecules (Chem. Sci. 8, 1482, 2017), while using no IPEA shift leads to a lowering of the excitation energies. Did the authors use an IPEA shift? Of course, such discussion can be moved to the SI.

4. Figure S1 needs some clarification:

- TDDFT is CAM-B3LYP?

- convoluted spectra correspond to ensembles from Wigner sampling rather than convoluting the FC excited states?

- Fig. S5 from Ref. 22 with the experimental gas-phase spectrum only provides data until 225 nm, while here the experimental spectrum extends to 200 nm? Did the authors include the theoretical results from Ref. 22 by accident?

5. It is stated, that the protocol from ref. 41 was used to select the initial excited states in the dynamics simulation which is based on defining an energy window from which to select the excited states based on their oscillator strength. In the discussion, it is mentioned that all states were started in the S6 state, in 116 out of 300 geometries. Did the authors choose an energy window (if so, which?) that happened to include only (adiabatic) S6 states, or did they select only S6 states to start the trajectories? How did some trajectories then end up in higher-lying states S7-S9 (Figure S6/Figure S8/Table S3)?

6. In relation to point 5: Assuming that CAM-B3LYP does show a systematic underestimation of excitation energies (see comment 3), the simulations corresponding to the experiments with excitation at 267 nm should be started at a lower wavelength. 267 nm/4.64 eV shifted by, e.g. 0.5 eV would start at 4.14 eV, which according to Table S1 would rather correspond to the S4/S5 states, so the dynamics should have been started there. Since both S4/S5 and S6 are of MLCT character and given the clear picture obtained from the dynamics simulation, I do not expect the different initial states to make a large difference. However, this point should be mentioned, so that one should be alert not to directly compare, e.g., time constants between the dynamics of the present study and experiments with 267-nm excitation.

7. p 6, l.156: it is stated that dissociation starts with i) a prominent elongation of the Fe-C bonds. Does this include all Fe-C bonds or only of the Fe-C bonds with the dissociating CO ligands (as in point ii)?

8. The period of the CO oscillators of 90 fs would correspond to a vibrational frequency of 370 cm⁻¹. Is there any mode in the ground state with that frequency that could also (partially?) be responsible for the observed oscillations? Would the observed behavior then be due to initial conditions in the ground state?

9. Figure 3a: The dynamically averaged potentials include data from all 110 singly dissociative trajectories. This data is dominated by complexes that lose their axial CO ligand (85 vs 15%). Does it make sense to average over all the data in one set? Would the potentials change if looking only at

the complexes that lose their equatorial CO ligand? The wiggles in all curves seem to appear at the same distances. Would these wiggles disappear if differentiating between axial and equatorial CO loss?

10. p 9, l.254: Can the authors specify how many trajectories undergo CO loss in each of the bursts?

11. p 11, l.296 states that "we can conclude that angular distortion modes are also vibrationally excited in the bound states". What does this mean? The classical picture for nuclei used in ab initio molecular dynamics does not have any vibrational states, so how do the authors conclude that vibrationally excited states play a role in the bound (electronic) states?

12. p11, l302: why did the authors choose to follow the S5 to its minimum? What about the S4?

13. p13, l368: it is stated that the oscillatory release of CO comes as a consequence from periodic transitions from the MLCT to the MC states. While the oscillatory release of CO has been clearly shown, the periodic transitions between MLCT and MC states have not really been demonstrated, i.e., this is only an assumption so far. The periodic transitions should be visible when analyzing the time evolution of the electronic states in a suitable way. This could be done, e.g, by combining the populations of the dissociative states (MC) and non-dissociative states (MLCT) in Figure S8. If the assumption is correct, this should also show stepwise population transfer between both sets. If it is not possible to characterize the adiabatic populations in Figure S8 clearly as diabatic dissociative states (MC) and non-dissociative states (MLCT), however, the authors could try to use the transition-density-analysis tools in SHARC based on the TheoDORE program (JCP 152, 084108 ,2020) which can easily distinguish between MC and MLCT states.

Reviewer #3 (Remarks to the Author):

In their manuscript entitled "Photoinduced bond oscillations in ironpentacarbonyl give delayed, synchronous bursts of carbonmonoxide release", A. Banerjee, M. Odelius and colleagues use computational methods to elucidate the pathways for CO release from Fe(CO)₅. The current methodological limitations of the inorganic photochemistry community dictate their choice of computational approach: since wave function methods in combination with excited state molecular dynamics are not accessible, Banerjee et al. resort to a TDDFT treatment of the electronic structure for the dissociation problem at hand. In this methodological choice and the way it is discussed,

presented and not fully reinforced by complementary calculations lies my biggest criticism of the manuscript.

Overall, I enjoyed reading this manuscript and believe that it will make an important contribution to the literature. The authors have analysed the data in detail and make useful connections to the literature, experimental data and established concepts. However, since the methods applied are at the brink of what is feasible currently with computational chemistry methods, I believe that a more in depth discussion of the methodological limitations would lift the results and enhance the impact of the interpretation.

The topic is timely and the general insights to be gained from a theoretical chemistry approach to the subject are of broad interest and hence suitable for Nature Communications. Given the main points of concern listed below, I recommend major revisions. As a note to the editors, I would expect that some of the required calculations take significant time.

(1) Methodological choices

1a TDDFT. The presentation of the TDDFT results is lacking. In Figure S1, the labelling does not allow the reader to understand immediately which density functional or active space, basis set and other technical settings were used.

In Figure S1, I do not find the agreement between experiment, TDDFT and NEVPT2 convincing. The authors describe this in the main text as "similar shapes", but given the lack of detail in these spectra this appears an overstatement. The TDDFT-S6 state is clearly much more intense than in the other two spectra printed. Intensities are not reported in Table S1, rendering a full and direct comparison of methods impossible. While some MOs are shown in Figure 1c, the character of the states is not reported here; from the reader's perspective it would be beneficial to have some descriptors, e.g. involved MO coefficients or transition or difference densities, available here rather than only in the latter part of the manuscript.

1b States involved. The authors restrict the ORCA/SHARC dynamics to singlet states and argue that this is motivated by experimental data. From the limited information given in the manuscript and the discussion in reference 10 (P. Wernet et al.) it is not clear whether the experimental data would be (a) sensitive enough to detect small quantities of very short-lived triplet species, (b) whether spin orbit coupled singlet/triplet states would be detectable, (c) whether `_excited_` state triplets can be detected. Since ref. 10 states "We unambiguously find that neither the transient intermediate $\text{Fe}(\text{CO})_4$ nor $\text{Fe}(\text{CO})_3$ on time scales up to 6 ps occurs in their triplet ground states. This validates the proposed singlet pathway for $\text{Fe}(\text{CO})_5$ photodissociation.", the authors should clarify this point.

To provide proof that triplet states are high in energy and therefore irrelevant for the dissociation path, the authors could spot check on the trajectories and scans with single point calculations. They

may consider including broken symmetry DFT calculations to capture open-shell singlets that may appear in the bond breaking region, unless they can put forward arguments that conceptually rule out triplet and open-shell singlet states.

1c Single- vs. multideterminant description. Describing dissociation with DFT and TDDFT suffers from well known problems due to the single determinant nature of the method. While the authors acknowledge this in the methods section and when describing the failed trajectories, the discussion does not go deep enough in my opinion. How can the authors be sure that the DFT/TDDFT electronic structures do not suffer from any problems before they fail? What criteria have they used to assess their validity? If any energetic or electronic structure uncertainties arise, how much would they affect the interpretation qualitatively and quantitatively? A detailed discussion will likely be too long for the main text, but would have its place in the SI of such a high level publication.

Perhaps the biggest methodological gap I view in this work is that since the TDDFT-based SHARC dynamics reveal a clear dissociation coordinate, why did the authors not follow up on this with wavefunction calculations along this coordinate? Even if the TDDFT had problems in the dissociation area, the wave function calculations in one dimension would provide a much clearer and complimentary picture of the dissociation process. To clarify this point, the authors could use wave function methods to check for multideterminantal character of the wavefunctions where prudent.

(3) Scans

In Figure S2, it is not clear how fine the scan is; the authors should revise these figures and include markers for the individual data points in the horizontal direction (e.g. on the PESs or as a separate horizontal line). The same applies to Figures S7 and S10.

Figure S2 also shows that the TDDFT- and NEVPT2-descriptions of the dissociative states differ. The authors should add a more detailed reasoning why they do not think that these differences matter. I would be more hesitant to treat the different crossing points of the dissociative states at (roughly) 1.85Å vs. 1.7Å and 2.15Å vs. 1.95Å as identical for the dissociation process.

(4) Dissociation criteria

The authors use as a dissociation criterion an Fe-C distance of 2.5 Å, while the PES scans show a significantly earlier crossing of the dissociative state(s) for the different methods. It is surprising that no electronic criterion is used, or even discussed why a purely distance-based choice was made. The authors should add a more detailed explanation here. How does the breathing mode with an amplitude of 0.3 Å and the equilibrium bond length of 1.8 Å fit in this picture – or phrased

differently, what happens (electronically or structurally) in the species with bond lengths between 2.1-2.5 Å?

(5) Electronic structure discussion

The authors provide some electronic structure discussion, but there are several cursory statements like "As discussed earlier in detail from different points of view, the photodissociation happens by a transfer from non-dissociative to dissociative adiabatic states, which are associated with diabatic MLCT and MC states at long Fe-C distances." that do not do justice to the amount of data and level of analysis the authors have shown. I would encourage the authors to elaborate on the electronic structure aspects wherever they see fit, as they have done for explaining the minority pathway.

(6) Discussion of changes in CO bonding

I may have overlooked this, but I did not see any discussion of changes in the CO bonding situation. This would be very valuable to experimental chemists since the CO stretching frequency is a sensitive probe of electronic structure and geometry, and can be monitored with high accuracy. Can the authors extract or extrapolate which changes they expect in the vibrational spectra along the dissociation process?

Responses to Reviewers' Comments.

Reviewer #1 (Remarks to the Author):

Banerjee et al. employed excited-state molecular dynamics to study the early dynamics of Fe(CO)₅ after excitation at 267 nm. The calculations were carried out using semi-classical simulations with SHARC at the TD-DFT level of theory. The calculations provide several detailed mechanistic insights into the early photophysics/photochemistry of Fe(CO)₅. Specifically, the calculations find, in support of earlier experiments, that a dark metal-centered state with anti-bonding Fe-CO character drives the dissociation of one CO from the ironpentacarbonyl complex. The calculations also predict excited-state driven oscillations in the axial CO release lifetime along with a rotation/pyramidalization mode that results in equatorial CO release.

Overall, the calculations appear to be carefully conducted and executed. My opinion is that this work is publishable. I have a few comments, mostly minor points:

Our Response: We appreciate the concise summary of the study and its main results. We also thank the reviewer for the careful reading and positive evaluation of our manuscript, and we will address the detailed comments below.

R#1 comment 1: Line 106: "Apart from the consistent underestimation of all excitation energies, we notice in Table S1 in the SI a close agreement between TDDFT and high level quantum chemistry, differing essentially only in the ordering of the quasi-degenerate bright $1A_2''$ and dark $1E'$ states, due to the different treatment of dynamical correlation, which earlier have been identified as being possibly involved in the excitation at 267 nm." —> I don't know if I would call the TDDFT and NEVPT2 results a "close agreement." Not only is there a systematic redshift of TDDFT calculations by as much as ~1 eV, but the ordering of first 4 states also differ. Clearly, the NEVPT2 results seem to be in better agreement with the experiments, although the NEVPT2 calculation quality will likely quickly deteriorate for higher excited states due to the 10,10 active space used. However, the good news is that the energy difference between the bright $1A_2''$ (MLCT) and dark $1E'$ (MC) states are consistent for TD-DFT and NEVPT2, and those are the states largely driving the dynamics.

Our Response:

We thank the reviewer for the critical comments on the details in the comparison and for noticing a key similarity, in the energy difference between bright $1A_2''$ (MLCT) and dark $1E'$ (MC) states, of the TDDFT and NEVPT2 potential energy surfaces. We have modified the text and the SI to incorporate a more elaborate discussion of TDDFT in relation to NEVPT2 and the key similarity highlighted by the reviewer. The TDDFT and NEVPT2 scans

as a function of Fe-C distance show that, although there are differences in the details, the overall PES shows similarities. The scan consistently shows from both TDDFT and NEVPT2 that as the Fe-CO bond distance increases the energy separation between the dissociative MC states and MLCT states increase, with the dissociative dynamics being confined to the MC states. We have included a detailed discussion on the comparison and validity of TDDFT by comparison to NEVPT2 in SI.

Explicit changes in the manuscript and the SI:

On page 5 of the manuscript, we have reformulated and added a comment on the energy difference in the potential energy curves between the excited MLCT state and the MC states, and similarity of the NEVPT2 and TDDFT results. Further down on page 5 with reference to the new Figure S4 in the SI, we also discuss the similarity between NEVPT2 and TDDFT potential energy surfaces along an example excited state trajectory.

On page 5 of the manuscript and in new Figure S5 and on page 7 in the SI, we also added a discussion of the character of the ground state and the validity of the TDDFT framework along the dissociation coordinate.

On page 2 in section A of the SI, we have added an extended discussion about the relation between the TDDFT and the NEVPT2 results, and about our methodological choice.

R#1 comment 2: On a related note: please include computed oscillator strengths for each state's transition from the ground state in Table S1. It will make it easier to compare Table S1 and Fig S1.

Our Response: We thank the reviewer for pointing out that quantitative information is useful to report. We have included oscillator strength in Table S1. We have also included new Figure S1 that includes the active MO used in the CASSCF computation and also the states character in terms of excitations among those orbitals. In the processes we also corrected typos in orbital notation.

Explicit changes in the manuscript and the SI:

On page 4 of the manuscript, we have added reference to the new Figure S1.

On page 2 in Table S1 of the SI, we have added information about the oscillator strength and refer to new Figure S1 for the characters of the states in terms of orbital transitions.

R#1 comment 3: Line 154: "the dissociation process (see Figure 2a) involves:

- i) a prominent elongation of the Fe-C bonds,
- ii) followed by preferential elongation of an axial Fe-C bonds, and
- iii) angle distortion from D_{3h} towards C_{4v} symmetry."

—> Despite how it is written, only point ii) can really be seen in Fig. 2a. Point i) isn't really clear in any of the figures (except in the SI) and point iii) is only clear later in Fig. 4.

Our Response: We thank the reviewer for noticing the need to clarify this point. We have made modifications to Fig2a.

Explicit changes in the manuscript and the SI:

On page 7 of the manuscript, we have added a comment on the angular distortion.

In Figure 2a and corresponding part of figure caption we now highlight active degrees of freedom in the dissociation process.

R#1 comment 4: On a related note, a few parts of the paper are difficult to navigate without referring to the SI.

4a) For example try reading this without referring to the SI at each sentence: "Based on this we did kinetic modelling (see Figure S9 and discussion in the SI for details) for the population dynamics. The fitted curves, of electronic population are overlaid with the simulated data as shown in Figure S8. The area under fitted curves were taken as a measure of lifetime of different states (since due to the presence of back reactions exponential fits was not an option) and was found to match well with the lifetime data from the simulation (see Figure S2). This also establishes that the dynamics can be accurately described in a simple model, in which hopping from one state predominantly happens to the adjacent adiabatic states."

Our Response: We thank the reviewer for pointing out that the presentation is in part difficult to follow. To remedy this problem, we have tried to improve the presentation by making the changes listed below.

Explicit changes in the manuscript and the SI:

On page 10 of the manuscript, to make the text more easy to read, a summary of the population dynamics is replacing the extended discussion in this paragraph, which in turn is moved to the SI.

On page 16 of the SI, we have now put the discussion from the main text, which contains numerous references to the SI itself.

4b) I'm not sure how to best address this, since I appreciate that there are figure limit in the journal and that not all technical figures can be moved into the main text. Maybe, instead, such details should be moved to the Methods section at the end? Alternatively, maybe Scheme 1 could be moved to the SI and some of the more important figures from the SI could be moved into the manuscript. This is just a suggestion.

Our Response: We humbly disagree with the reviewer here and we find Scheme 1 valuable to keep in the manuscript, instead details on population dynamics and kinetic modelling are now only briefly discussed the main text, and is mainly relegated that to the SI. The major interesting dynamics like that of second burst and third burst are

included in the main text, and, Scheme 1 is an attempt to conceptually summarize the entirety of the study.

Explicit changes in the manuscript and the SI are those mention in response to the previous point

See changes under **R#1 comment 4 4a)**

R#1 comment 5: Instead of referring to the states as S1, S2, S3, etc., in several places in the manuscript it would be better to refer to the electronic character (MLCT, MC) or by symmetry labels where possible/appropriate. The reason is that the ordering of these states changes with geometry, computational method, etc. and so such labels are not really meaningful except to refer to the ordering of the states at the equilibrium geometry for a particular method. For example, near the equilibrium geometry S1-S4 are not dissociative, they only become dissociative once the CO metal bonds are elongated as can be seen in the diabatic surfaces in Fig. S7.

Our Response: We thank the reviewer for this idea of how to improve the presentation. We have put this into consideration and changed the notation where ever we found it is possible, to facilitate for the reader and conform with the reviewer's comment. The challenge is that our population analysis is restricted to adiabatic state, S₀-S₉, and we want to maintain a precise language.

Explicit changes in the manuscript and the SI:

On page 8 of the manuscript, we have added the character of the S₈ and S₃ states.

On pages 12-13 of the manuscript, we have added the character of the S₄ and S₅ states.

On page 17 of the manuscript, we discuss the character of the S₆ state.

On page 8 of the manuscript, in the figure caption of Fig3 we have highlight the characters of the non-dissociative and dissociative states.

R#1 comment 6: Related to the point above: the authors sample 300 initial conditions and only consider 116 which were excited into the "S6" state. If I understood this correctly, it means that the authors only chose states where S6 has 1A2" character (or a large oscillator strength) and excited into those, discarding the rest. In this case, they could be discarding states where other states, e.g., S7 or S5, have the 1A2" character due to some geometric distortions. In this way, the authors exclude trajectories that could potentially be interesting (where the molecule is excited along modes that reorder state energies) and bias the results of the dynamics.

I don't expect this will qualitatively affect the results of the calculations, but it could very well skew the relative yield of trajectories that give axial vs. equatorial dissociation or double dissociation.

Our Response: We are grateful to the reviewer for pointing out this potential problem in sampling. We have chosen S_6 trajectories, and these have MLCT character based on the large oscillator strength. We acknowledge that we have neglected the trajectories which could have been S_7 or S_5 could become bright and have MLCT character. This is a limitation of our study but we do not think this can change the relative yield of the trajectories as the shape of the potentials are similar for S_5 , S_6 and S_7 . Also, it should be noted that trajectories eventually also go into S_5 and S_7 during the dynamics and that at early times near the Frank-Condon region. We have added a section on this in the methodology section.

Explicit changes in the manuscript and the SI:

On page 15 of the manuscript, we have added a discussion of this the choice of initial conditions for the simulations.

R#1 comment 7: I caught a few typos while reading the manuscript (likely, there may be a few more, just reporting those I noticed):

- line 26: there is no need for the comma after bonds.
- cam-B3LYP should be CAM-B3LYP (capital letters).
- line 246: "The fitted curves, of ..." → The comma is not needed.
- line 248: "exponential fits was" → "exponential fits were"
- line 379: anit-bonding → anti-bonding.
- line 409: "We also checked ... the CASPT2(12,12) geometry." → some commas are needed here, or split into two sentences.
- line 424: stats → states

Our Response: We are grateful to the reviewer for the careful reading and apologize to not detecting these problems ourselves. We have now proof read the manuscript and SI. All these typos and a few additional typos have been corrected following the reviewer's comment at all the relevant places.

Explicit changes in the manuscript and the SI:

Specific comments are carefully reviewed and corrections are made in text and also in SI.

Reviewer #2 (Remarks to the Author):

R#2 comment A: The present manuscript of Banerjee et al. presents simulations of the excited-state dynamics of the pentacarbonyl-iron complex in gas phase after UV excitation. Excited-state dynamics simulations of transition-metal complexes are still very challenging. Most past studies used quantum dynamics methods, that were, however, limited to only consider a few a priori selected degrees of freedom to describe the excited-state reactions. Only in the last few years, full-dimensional simulations using ab

initio molecular dynamics methods such as used in this work have become possible for transition metal complexes, and the examples of these applications are still rare. Due to the small size and simple structure, pentacarbonyl-iron is a good test system for both experimental and theoretical methods. However, despite this, its excited-state dynamics involving CO dissociation after UV excitation are still not well understood. Thus, I think, the present study can be interesting to a wide audience of theoretical and experimental chemists and could be published in Nature Communications.

Author Response: We thank the reviewer for valuing our efforts on studying the photodissociation on iron pentacarbonyl, and confirming the broad interest for the readership of Nature Communications. We notice the comment on the challenge in simulating photochemical processes in transition metal complexes, like $\text{Fe}(\text{CO})_5$. This has forced us to adopt a pragmatic approach to the choice of quantum chemical method TDDFT, evaluated along certain coordinates against high-level calculations NEVPT2.

R#2 comment B: However, I have a concern regarding the role of triplet states in the dynamics, that should be clarified to validate the conclusions drawn in the study. In the present simulations, it was assumed that the CO dissociation takes place entirely in electronic singlet states. This assumption is based on earlier experimental studies (Ref. 9 and 10 in the manuscript) which suggest that the first CO dissociation occurs via a singlet pathway and that only later triplet states may become populated in the dissociated product. Previous simulations using ab initio MD or quantum dynamics have shown, however, that in iron complexes, intersystem crossing can occur on the (sub-) 100 fs time scale (e.g., JPCL 7, 2009, 2016; Inorg. Chem. 59, 14666, 2020; similar studies exist also for other 3d metal complexes). As this timescale is similar to that observed for the CO dissociation in the present simulations, one may expect that intersystem crossing could also compete with CO dissociation in the singlet manifold in pentacarbonyl-iron. Thus, I think the authors should demonstrate that their assumption to include only singlet states is justified.

Our Response: We thank the reader for bringing up this important point in our study. Firstly, three independent studies have, based on three different spectroscopic techniques, ruled out the presence of intersystem crossing (ISC) and triplet pathway in the photodissociation of $\text{Fe}(\text{CO})_5$ in the gas phase. Trushin and co-workers (J. Phys. Chem. A 2000, 104, 10, 1997–2006) have shown in their work that singlet pathway to be operative with no ISC. They have, based on computation of time-constants for the dissociation process, excluded the possibility of ISC happening, arguing that the photodissociation happens much faster than the ISC. Additionally, Wernet and co-workers (The Journal of Chemical Physics 488 146, 211103 (2017).) have corroborated the above claim based on the high-temporal resolution gas phase XPS studies. XPS is very sensitive to the electronic structure of the species, not least to the multiplicity due to strong spin-orbit coupling in the final states of the XPS probe, and would in principle have picked up the formation of triplet species or if the triplet pathway had a major involvement. They have confirmed the

absence of any triplet involvement up to 6ps which is beyond the temporal regime of our present studies. Recently using time resolved IR experiments Ramasesha and co-workers have shown that the ISC channels are only operative in the $\text{Fe}(\text{CO})_4$ moiety after 15 ps timescale (J. Chem. Phys. 154, 134308 (2021)). Thus, gathering from three independent experimental studies that triplet pathways are not operative it is pragmatic for us to exclude the triplet states in the study, and investigation of possible involvement of triplet states for less probable/minor pathways will have to await further investigations and improvements in simulations capabilities.

Nevertheless, in an attempt to find a justification for the singlet pathway and the absence of triplet states in the excited state dynamics following photo-excitation of iron pentacarbonyl in gas phase, we have made Fe-CO scans for both singlets and triplets involved at the NEVPT2 level of theory which is robust in its treatment of triplet and computation of SOC elements. (We also made a corresponding scan at the TDDFT level). The triplet manifold is an analogue of the singlet manifold, i.e., it comprises of four ^3MC states which are dissociative in nature and followed by these are the $^3\text{MLCT}$ bound states. In the new Figure S6 in SI we show the NEVPT2 PES of the triplet (black lines) and singlet (blue lines). The two papers suggested by the reviewer (JPCL 7, 2009, 2016; and Inorg. Chem. 59, 14666, 2020) describe ISC happening between bound states having similar shapes (non-dissociative and hence described by LVC model). The crossing between the $^3\text{MLCT}(\text{bound})$ and $^1\text{MLCT}(\text{bound})$ states (denoted by green circle) are energetically much higher as compared to the crossing between $^1\text{MLCT}(\text{bound})$ states and the $^1\text{MC}(\text{dissociative})$ states for the present case. Thus the rate for transition from the $^1\text{MLCT}(\text{bound})$ states to the $^1\text{MC}(\text{dissociative})$ state will be faster as compared to the ISC rate for the present system. The black circles in Figure S6 denotes the crossing between the $^1\text{MC}(\text{dissociative})$ state and the $^3\text{MLCT}(\text{bound})$ states. The ^3MC states which are also dissociative in nature are well separated from the singlet manifold. Now the process involves excitation to a MLCT state followed by decay to dissociative ^1MC states followed by fast dissociation. The internal conversion within the singlet manifold to the dissociative ^1MC state happens before the region where ISC could occur and by the time the system encounters the ISC region (highlighted by the black circle), the system is already in the process of a fast dissociation on a repulsive surface. Thus, the system likely spends too little time in that region for ISC to effectuate, or in other words the system has too large a nuclear velocity, which is an important factor in ISC rate, for ISC pathway to be competitive. Though the spin-orbit coupling (SOC) between a ^1MC and $^3\text{MLCT}$ states is expected to be low, when computed we found it to the order of $\sim 220 \text{ cm}^{-1}$, at NEVPT2 level of theory. Considering the timescale connected to the SOC is $\sim 75 \text{ fs}$ under ideal conditions, given the system only spends in the resonant region due to the fact that singlet states are dissociative and triplet states are bound, we would assume that the branching ration to the triplet state is rather low. This provides a justification for the experimentally observed singlet pathway. Alternatively, the nuclear wavepacket overlap, which is an important aspect in the ISC process, can also assumed to be low given the different shape of the

crossing singlet(dissociative) 1MC states and 3MLCT triplet(bound) PES. This is again in stark difference to the two papers mentioned above (JPCL 7, 2009, 2016; and Inorg. Chem. 59, 14666, 2020). This discussion has been included in the SI and appropriate changes indicating towards this discussion in SI is made in the manuscript along with citation these two important references as mentioned by the referee, which highlight another important aspect of iron-photochemistry.

Explicit changes in the manuscript and the SI:

On page 2 of the manuscript, we have added a sentence referring also to the work of Ramasesha and co-workers [newref11].

On page 6 of the manuscript, we have added discussion of the triplet states and the intersystem crossing in this and other systems.

On page 8 in section B of the SI, we have also added an extended discussion of the influence of the triplet states together with potential energy surfaces in new Figure S6.

R#2 comment 1: In addition, I have a series of minor concerns that should also be addressed (possibly also in the SI). The documentation of the computational details needs to be more specific in several places:

- specify the version of the MOLPRO package
- details on parameters in DFT/TDDFT calculations: grid size, convergence parameters, etc., use of Tamm-Dancoff approximation (default in ORCA)?
- specify the default parameters in the SHARC simulations, e.g., decoherence correction, energy/momentum rescaling, etc... that are missing in the current computational details. Such defaults can change over time and should, thus, be documented in the studies where they are used.
- specify CASSCF/CASPT2/NEVPT2 variants: state-averaged CASSCF? Multi-state CASPT2? PC-NEVPT2? Did the authors use any shifts in the CASPT2 calculations? Also, a figure of the active orbitals would be welcome for the interested reader.

Our Response: We are grateful for the careful reading of the manuscript, and have tried to improve the presentation. The details of computation are further enhanced following the point-by-point suggestion of the reviewer and are provided in the computational details in the Methods section.

Explicit changes in the manuscript and the SI:

On page 16 of the manuscript, we have specified the version of MOLPRO.

On page 17-18 in sections B and C of the manuscript, we have specified further computational details about the excited state calculations and the dynamic simulations.

R#2 comment 2: Figure 1: the relative intensity of the convoluted TDDFT equilibrium

curve does not seem to match with the four shown transitions, where the second (higher-energy) absorption band has almost twice the intensity than the first one, while both bands seem to contain states with equal oscillator strength. How was this curve calculated?

Our Response: The reason that the higher energy (lower wavelength) has almost twice that intensity in the convoluted spectra is because there are two degenerate states S_{19} and S_{20} , which are of $^1E'$ state) but the one at lower energy is the single state S_6 corresponding to the $^1A''_2$ transition. When convoluting the degenerate S_{19} and S_{20} contribute twice to the height of the convoluted spectra, and in the stick spectrum there is only a single line. The curves are obtained by convoluting the sticks with a Lorentzian broadening of 0.8 eV using `orca_mapsc` script given with ORCA package.

Explicit changes in the manuscript and the SI:

On page 4 in SI the caption of Figure S2 of the SI, we have described the convolution scheme.

R#2 comment 3: p 4, l.111: the authors mention the close agreement between TDDFT results and high-level quantum chemistry results, although the excitation energies between CAM-B3LYP and NEVPT2 results differ by 0.7-1.4 eV in Table S1, which cannot really be considered good agreement anymore. Compared to the experimental absorption spectrum, it seems that CAM-B3LYP underestimates the excitation energies, while NEVPT2 overestimates the excitation energies. So, given both errors were systematic, this could explain the large differences and still justify the use of either method. However, I think the authors should try to demonstrate this. Are similar large differences between TDDFT and CAS methods for other 3d-transition metal carbonyl complexes known? NEVPT2 has been shown to give similar results as CASPT2 when using an IPEA shift of 0.25 au, and both tend to overestimate experimental excitation energies, at least for organic molecules (Chem. Sci. 8, 1482, 2017), while using no IPEA shift leads to a lowering of the excitation energies. Did the authors use an IPEA shift? Of course, such discussion can be moved to the SI.

Our Response: In the discussion of disagreement and agreement, we refer back to the response to **R#1 comment 1**, and how we could argue to claim that even a NEVPT2 treatment could give a similar qualitative behavior. The methodological choices have been discussed in detail following the questionnaire from **Reviewer 3**. We would like to point that higher lying excited states are often not described accurately by NEVPT2/CASPT2 level of theory, mainly because of the fact that these operate on limited active spaces, which is 10e,10o for our case. Our TDDFT excitation energies are in-fact very close to the excitation energies obtained by Daniel and co-workers (Chemical Physics Letters 302, 489-494 (1999) using MRCISD done on CASSCF(8e,16o) reference computation (see table below).

States	TDDFT(CAM-B3LYP) / eV from Table S1	MRCI(8e/16o) / eV	MRCI-TDDFT / eV	CASPT2(no IPEA) / eV	CASPT2 / eV from Table S1
¹ A ₁ "	3.89	4.12	0.23	4.40	4.88
¹ E'	3.97	3.55	-0.42	4.21	4.48
¹ E"	4.17	4.55	0.38	4.593	5.12
¹ A ₂ "	4.4	4.59	0.19	4.533	5.11
¹ A ₁ '	4.53	-	-	-	-
¹ E'	4.66	4.87	0.21	5.25	6.33

We thank the reviewer for pointing out that we should also have a discussion of the CASPT2 and NEVPT2 comparison, and we have elaborated on the matter in the SI. We use FIC-CASPT2 as implemented in ORCA 4.2 with imaginary shift of 0.2 and IPEA shift of 0.25. CASPT2 (no IPEA) gives excitation values lower than that when using IPEA of 0.25, was rightly pointed out by the reviewer and shown in manuscript actually gives value which are much closer to the TDDFT value, but the results get worse for higher excited states which are due to limited nature of the active space.

Explicit changes in the manuscript and the SI:

First of all we refer to response to **R#1 comment 1**.

On page 17-18 in section C of the manuscript, we have specified further computational details about the excited state calculations.

On page 2 of the SI, we have added a section on the comparison of the NEVPT2/CASPT2 and TDDFT calculations. The CASPT2 with NO-IPEA values has been added in the Table S1 and a short discussing along with citation of (Chem. Sci. 8, 1482, 2017) has been included in page 5 of the main manuscript. We also discuss the overestimation of excitation energies from NEVPT2 and CASPT2(IPEA: 0.25) following the suggestion of the reviewer. The MRCI(8e/16o) data from Daniel and co-workers have also been added in the Table S1 in SI.

R#2 comment 4: Figure S1 needs some clarification:

- TDDFT is CAM-B3LYP?
- convoluted spectra correspond to ensembles from Wigner sampling rather than convoluting the FC excited states?
- Fig. S5 from Ref. 22 with the experimental gas-phase spectrum only provides data until

225 nm, while here the experimental spectrum extends to 200 nm? Did the authors include the theoretical results from Ref. 22 by accident?

Our Response: We are grateful for the remarks about flaws in the presentation, and we have tried to make suitable changes. But the Figure S1 spectra shows spectra of convoluted sticks spectra obtained by excited state computation of the optimized GS geometry of Fe(CO)₅. We have now extracted the experimental data from a suitable reference and reproduce that in Figure 1 and Figure S2

Explicit changes in the manuscript and the SI:

On page 24, Figure 1 has been updated with the correct experiment spectrum and the proper reference is used. The reference is also properly cited on page 4.

On page 4 of the SI, we have also made changes to Figure S2(old Figure S1) and its caption, also in other places we have tried to adopt a clear notation.

R#2 comment 5: It is stated, that the protocol from ref. 41 was used to select the initial excited states in the dynamics simulation which is based on defining an energy window from which to select the excited states based on their oscillator strength. In the discussion, it is mentioned that all states were started in the S₆ state, in 116 out of 300 geometries. Did the authors choose an energy window (if so, which?) that happened to include only (adiabatic) S₆ states, or did they select only S₆ states to start the trajectories? How did some trajectories then end up in higher-lying states S₇-S₉ (Figure S6/Figure S8/Table S3)?

Our Response: We acknowledge the importance of clarifying this in the presentation, and thank the reviewer for the comment. We choose only those trajectories which has bright S₆ state, i.e, where S₆ state corresponds to a MLCT state. As noted in reply to the reviewer 1 on page 5 above, we have restricted the sampling in this way.

Now during dynamics, we observed that the system hopped on to higher state i.e. S_7 - S_9 . We think that this is due to fact that as the system relaxed the potential energy that the system has is enough to take the system to a higher state in that geometry. Notice that in these non-adiabatic transitions the state character could be preserved. This is something that we think is common and can found throughout the literature and we show it schematically in the left figure below.

Scheme: Schematic figure showing hypothetical PES where trajectories can go up to higher adiabatic state or remain in the same.

Explicit changes in the manuscript and the SI:

On page 15 of the manuscript, we have added a discussion of this the choice of initial conditions for the simulations.

On page 16 in SI, we added a discussion of the populations of the S_7 - S_9 states in newly introduced section III.

R#2 comment 6: In relation to point 5: Assuming that CAM-B3LYP does show a systematic underestimation of excitation energies (see comment 3), the simulations corresponding to the experiments with excitation at 267 nm should be started at a lower wavelength. 267 nm/4.64 eV shifted by, e.g. 0.5 eV would start at 4.14 eV, which according to Table S1 would rather correspond to the S_4 / S_5 states, so the dynamics should have been started there. Since both S_4 / S_5 and S_6 are of MLCT character and given the clear picture obtained from the dynamics simulation, I do not expect the

different initial states to make a large difference. However, this point should be mentioned, so that one should be alert not to directly compare, e.g., time constants between the dynamics of the present study and experiments with 267-nm excitation.

Our Response: We acknowledge the importance of this aspect and conclude that this is an aspect that should be mentioned more clearly. We resort to previous studies from Trushin and co-workers, which states that the excitation happens to the bright 1A_2 state (MLCT state). We agree that our work here is limited in the fact that we have not considered excitation to the other MLCT states, which can get populated when being hit by 267 nm light, which also has a broadening. We have included a discussion and also have presented the limitation of the study when it comes to direct comparison to the experiment.

Explicit changes in the manuscript and the SI:

On page 15 of the manuscript, we have added a paragraph discussing the initially excited states.

R#2 comment 7: p 6, l.156: it is stated that dissociation starts with i) a prominent elongation of the Fe-C bonds. Does this include all Fe-C bonds or only of the Fe-C bonds with the dissociating CO ligands (as in point ii)?

Our Response: As shown in Figure S9 all Fe-C bonds have been elongated as the vibrations happen about a distance longer than the equilibrium bond distance. To clarify this point, we have also changed the Schematic representation in the Figure 2a in Main Manuscript.

Explicit changes in the manuscript and the SI:

In Figure 2a the schematics is improved and the caption is modified.

R#2 comment 8: The period of the CO oscillators of 90 fs would correspond to a vibrational frequency of 370 cm^{-1} . Is there any mode in the ground state with that frequency that could also (partially?) be responsible for the observed oscillations? Would the observed behavior then be due to initial conditions in the ground state?

Our Response: We thank the reviewer for the inspiration to investigate the normal modes of relevant species more carefully. The Fe-O oscillation with the 90 fs period occurs in a MLCT state before transition to dissociative MC states. Hence, we find that it is natural to investigate not only the vibrational modes in the ground state, but in an MLCT state. We hence performed vibrational analyses at both the $S_0(\text{GS})$ and $S_5(\text{MLCT})$ geometries, and also in the S_5 state at the S_0 geometry even though it is not a minimum.

We identified the normal modes which correspond to the Fe-C bond vibration in the S_5 state which is the lowest of the MLCT state and which remains MLCT across the potential energy surface. We see that indeed a symmetric mode exist at 363 cm^{-1} and can be responsible for the oscillation in the MLCT bound states. Thus, the burst phenomenon can be traced back to this normal mode in the MLCT states.

Explicit changes in the manuscript and the SI:

On page 13 of the manuscript with reference to new Table S4 at page 19 in SI, we discuss the possible relation between the symmetric Fe-C stretch in the S_5 state and the oscillation period in the Fe-C bonds in the excited state dynamics. We also discuss the the response in the CO vibrations.

On page 20 in Table S4 of the SI, we report on the vibrational Fe-C stretch and C-O stretch modes of S_0 and S_5 state.

R#2 comment 9: Figure 3a: The dynamically averaged potentials include data from all 110 singly dissociative trajectories. This data is dominated by complexes that lose their axial CO ligand (85 vs 15%). Does it make sense to average over all the data in one set? Would the potentials change if looking only at the complexes that lose their equatorial CO ligand? The wiggles in all curves seem to appear at the same distances. Would these wiggles disappear if differentiating between axial and equatorial CO loss?

Our Response: We computed the dynamically averaged PES separately for the 16 cases of equatorial dissociation and the 94 cases of axial dissociation. We found the shapes of all these to be similar with 94 axial cases more resembling that of the total set of singly dissociating trajectories. This could be due to the simple fact that the axial case is the major contributor, but no stark difference could be seen for the case of equatorial dissociation. However, we found the wiggles in case equatorial to a bit more pronounced which could be due to the rather limited number of trajectories that we have.

Explicit changes in the manuscript and the SI:

On page 9 of the manuscript, we have added a discussion on the above response. We also moved the sentence on the NaI analogue to page 9.

On page 15 of the SI, we have added Figure S12 to include the dynamically averaged PES for the case of 94 axial dissociations and 16 equatorial dissociations.

R#2 comment 10: p 9, l.254: Can the authors specify how many trajectories undergo CO loss in each of the bursts?

Our Response: Yes, we agree with the reviewer that it is can be of interest to mention this information in the presentation.

Explicit changes in the manuscript and the SI:

On page 10 of the manuscript, we have specified in parenthesis the number of trajectories in the first (27) and second (40) burst of axial CO ligands.

R#2 comment 11: p 11, l.296 states that “we can conclude that angular distortion modes are also vibrationally excited in the bound states”. What does this mean? The classical picture for nuclei used in ab initio molecular dynamics does not have any vibrational states, so how do the authors conclude that vibrationally excited states play a role in the bound (electronic) states?

Our Response: We apologize for the loose language and we have tried to reformulate these results in the presentation. The excitation imparts the molecule with a potential energy, and places it in a region of the state which is not the minima. The molecule then distorts owing to the energy being channelled to different degrees of freedom. Thus, the energy causes larger amplitude of oscillation/distortion which corresponds to a vibrationally excited state and thus in other words, different degrees of freedom are vibrationally excited. Though our work relies on the classical dynamics a quantum analogue can be drawn especially in the absence of the tunnelling effects, which are negligible here.

Explicit changes in the manuscript and the SI:

On page 11 of the manuscript, we have reformulated the sentence about angular distortions.

R#2 comment 12: p11, l302: why did the authors choose to follow the S5 to its minimum? What about t the S4?

Our Response: Since the adiabatic S₄ state is a dissociative state, and is furthermore a MC state for most part of the PES. S₅ on the contrary is clearly a bound state and is MLCT state all throughout the PES. In principle we could have also done S₆, S₇ etc (other MLCT states). However, we choose S₅ as the lowest of these clearly MLCT state and having a bound potential. This was just a proof on principle that angular distortions happen in bound states and will also be there in other bound states. We have mentioned this also in the modified text.

Explicit changes in the manuscript and the SI:

On page 12 of the manuscript, we have added a sentence describing the optimization of the S5 state.

R#2 comment 13: p13, l368: it is stated that the oscillatory release of CO comes as a consequence from periodic transitions from the MLCT to the MC states. While the oscillatory release of CO has been clearly shown, the periodic transitions between MLCT and MC states have not really been demonstrated, i.e., this is only an assumption so far. The periodic transitions should be visible when analyzing the time evolution of the electronic states in a suitable way. This could be done, e.g, by combining the populations of the dissociative states (MC) and non-dissociative states (MLCT) in Figure S8. If the assumption is correct, this should also show stepwise population transfer between both sets. If it is not possible to characterize the adiabatic populations in Figure S8 clearly as diabatic dissociative states (MC) and non-dissociative states (MLCT), however, the authors could try to use the transition-density-analysis tools in SHARC based on the TheoDORÉ program (JCP 152, 084108 ,2020) which can easily distinguish between MC and MLCT states.

Our Response: We completely agree with the reviewer on this. The periodic transition from MC to MLCT states is not an assumption, and is deduced exactly in the way that the reviewer points, Its clearly seen from Figure 3(b and c) that the transfer of population from the MLCT(non-dissociative states) to MC(dissociative states) for the first and second happen periodically and the later have clear lag period. However, we did add the dissociative states and non-dissociative states and plotted them in Figure 3(b) and (c). We do see that indeed the transfer is step wise as seen from Figure 3b and c. The characterization of states is also seen by looking at the characters at elongated Fe-C distances, as seen from Figure S17. This is also discussed in the text. Unfortunately, we didn't know the use of TheoDORÉ when the simulations were done. We will keep this point in our future endeavours.

Reviewer #3 (Remarks to the Author):

R#3 comment A: In their manuscript entitled "Photoinduced bond oscillations in ironpentacarbonyl give delayed, synchronous bursts of carbonmonoxide release", A. Banerjee, M. Odellius and colleagues use computational methods to elucidate the pathways for CO release from Fe(CO)₅. The current methodological limitations of the inorganic photochemistry community dictate their choice of computational approach: since wave function methods in combination with excited state molecular dynamics are not accessible, Banerjee et al. resort to a TDDFT treatment of the electronic structure for the dissociation problem at hand. In this methodological choice and the way it is discussed, presented and not fully reinforced by complementary calculations lies my biggest criticism of the manuscript.

Our Response:

We completely understand and agree with the point that the reviewer raises. Above in response to R#1 comment and R#2 comment 5, we have discussed the evaluation and limitations of TDDFT for studying dissociative dynamics in $\text{Fe}(\text{CO})_5$ and here we elaborate further. We also refer to the corresponding changes we have made in the presentation in the main text and SI.

As is pointed out by the reviewer later, we are operating “at the brink of what is feasible” (citation from reviewer #3) from the electronic structure theories that can be used when carrying out excited state MD computations, there is very little room for us right now to do more correctly. $\text{Fe}(\text{CO})_5$ is a nice close shell system with a singlet ground state (GS), and hence conceptually TDDFT is applicable to it in dealing with the excited states. However, $\text{Fe}(\text{CO})_4$ has a triplet GS (though in the experiment it not observed) and thus with such systems the singlet GS often has a multideterminant character, (for example O_2). We are studying the dynamics from $\text{Fe}(\text{CO})_5$ to $\text{Fe}(\text{CO})_4$ and hence operating in a fuzzy zone when it comes to the extent the dynamics is correct. The limitations of TDDFT are broadly two-fold; one, when the GS has multideterminant wavefunction and secondly in dealing with double excitations. $\text{Fe}(\text{CO})_4$ suffers from the first limitation and hence excluded from our study. When it comes to the second reason the excited states of $\text{Fe}(\text{CO})_5$ are single excitation ones as confirmed from CASSCF computation, and hence TDDFT is applicable. Moreover, TDDFT has been used earlier in dealing with the dynamics of $\text{Cr}(\text{CO})_6$ (a close enough system) by Barbatti and co-workers (The Journal of Chemical Physics, 134, 164305 (2011)).

Additionally, we have taken one of the trajectories and have conducted NEVPT2 computations on those geometries taken every 5fs. Following the computation, the TDDFT and NEVPT2 energies were plotted as a function of Fe-C distance, as shown in the Figure S4. We see from the plot below the necessary similarities and the difference. The similarities from the two theories include the separation between the MLCT states and the MC states increases. This also confirms the picture that dissociation happens in the MC states. Obviously, the slope for NEVPT2 as revealed is a bit steeper and hence the dynamics at NEVPT2 level of theory could give a faster dissociation. Now one important disagreement that we clearly see from the Figure is that as Fe-C distance increases the disagreement between the TDDFT and NEVPT2 increase when it comes to the MLCT states. This is primarily due to fact that TDDFT fails for $\text{Fe}(\text{CO})_4$ as discussed earlier. However, as the disagreement is mainly more pronounced in the MLCT state whereas most of the dynamics happens on the MC states we are kind of safe except for at very large distance.

Explicit changes in the manuscript and the SI:

On page 5 of the manuscript, we have added a discussion on the above point.

On page 2 of the SI, we have added a discussion in section IA. Figures S4, S5, and S6 have been added including captions, and references in the main manuscript.

Overall, I enjoyed reading this manuscript and believe that it will make an important contribution to the literature. The authors have analysed the data in detail and make useful connections to the literature, experimental data and established concepts. However, since the methods applied are at the brink of what is feasible currently with computational chemistry methods, I believe that a more in depth discussion of the methodological limitations would lift the results and enhance the impact of the interpretation.

The topic is timely and the general insights to be gained from a theoretical chemistry approach to the subject are of broad interest and hence suitable for Nature Communications. Given the main points of concern listed below, I recommend major revisions. As a note to the editors, I would expect that some of the required calculations take significant time.

R#3 comment 1: Methodological choices

1a TDDFT. The presentation of the TDDFT results is lacking. In Figure S1, the labelling does not allow the reader to understand immediately which density functional or active space, basis set and other technical settings were used.

In Figure S1, I do not find the agreement between experiment, TDDFT and NEVPT2 convincing. The authors describe this in the main text as "similar shapes", but given the lack of detail in these spectra this appears an overstatement. The TDDFT-S6 state is clearly much more intense than in the other two spectra printed. Intensities are not reported in Table S1, rendering a full and direct comparison of methods impossible. While some MOs are shown in Figure 1c, the character of the states is not reported here; from the reader's perspective it would be beneficial to have some descriptors, e.g. involved MO coefficients or transition or difference densities, available here rather than only in the latter part of the manuscript.

Author Response: We acknowledge the useful comments of the reviewer, and we have tried to improve the presentation by giving more details on the electronic states.

TDDFT and NEVPT2 are two completely different theories and hence the difference between them is very well expected, and that too for a transition metal system. The accuracy of TDDFT and NEVPT2 spectra when compared to the experimental spectra are similar but in opposite direction, i.e., red shifted in TDDFT and blue shifted in NEVPT2. NEVPT2 and CASPT2 tend to overestimate the excitation energies, and CASPT2 can correct it when done without IPEA(0.0) correction, as shown in Ref.(Chem. Sci. 8, 1482, 2017) and pointed out by Reviewer 2. This point was addressed in reply to **R#2 comment 3**.

Now errors in vertical excitation energies estimated by NEVPT2 and CASPT2 could also be due to the use of a rather limited active space and the difference in accounting for

the correlation energy of the ground state wrt the excited state, which results in error in estimation of the vertical excitation energy. We show in Table above in reply to **R#2 comment 3**, how MRCI(8e,16o) computed by C. Daniel et al conforms to this point.

We are however not interested in the energy wrt to the GS during the dynamics itself. Rather the excited state dynamics is governed by the relative energetics between the excited states. The relative energy difference between excited states, i.e., for example between the bright $^1A_2''$ (MLCT) and dark $^1E'$ (MC) states are consistent for TD-DFT and NEVPT2, and those are the states largely driving the dynamics (as also positively pointed out by **Reviewer 1**). Though there is reordering of the states among themselves also, if we look carefully at Table S1, the ordering is among those states which are quasi-degenerate by less than 0.5 eV, and that is most likely the result of the difference in treatment of correlation energy in TDDFT vs NEVPT2(10,10) level of theory. Moreover, as the bond is broken and is shown in Figure S2, the shape of the PES essentially have similar shape and though the details are different the overall picture is same. And at present point we are limited to TDDFT, but based on the present results, we are evaluating the possibility to further investigate the grid-based quantum dynamics in reduced dimension on a higher level of theory (e.g., NEVPT2). Its noteworthy to mention here that when we began the excited state MD simulation, we did so at the CASSCF level of theory (i.e., from the S_8 ($^1A_2''$) (MLCT) state and that resulted in dissociation of double CO dissociation, with direct formation of $Fe(CO)_3$. This is in direct contradiction to what is experimentally observed, indicating that dynamically correlation is important. Thus, TDDFT was the method of choice for our study.

However, we have taken into account the reviewer's suggestion we have made the changes in Table S1 and included the oscillator strength, and detailed descriptors of the excited states along with involved MO-coefficients. Additionally, a complementary picture of the active orbitals is also given.

Explicit changes in the manuscript and the SI:

On page 4 of the manuscript, we have added reference to the New Figure S1.

On page 2 in Table S1 of the SI, we have added information about the oscillator strength and fortified it with newer level of theories and referred to New Figure S1 for the characters of the states in terms of orbital transitions.

Other changes follow from previous replies to **R#1 comment 1 and R#2 comment 3**.

1b States involved. The authors restrict the ORCA/SHARC dynamics to singlet states and argue that this is motivated by experimental data. From the limited information given in the manuscript and the discussion in reference 10 (P. Wernet et al.) it is not clear whether the experimental data would be (a) sensitive enough to detect small quantities of very short-lived triplet species, (b) whether spin orbit coupled singlet/triplet states would be detectable, (c) whether _excited_ state triplets can be detected. Since ref. 10 states "We

unambiguously find that neither the transient intermediate $\text{Fe}(\text{CO})_4$ nor $\text{Fe}(\text{CO})_3$ on time scales up to 6 ps occurs in their triplet ground states. This validates the proposed singlet pathway for $\text{Fe}(\text{CO})_5$ photodissociation.", the authors should clarify this point.

To provide proof that triplet states are high in energy and therefore irrelevant for the dissociation path, the authors could spot check on the trajectories and scans with single point calculations. They may consider including broken symmetry DFT calculations to capture open-shell singlets that may appear in the bond breaking region, unless they can put forward arguments that conceptually rule out triplet and open-shell singlet states.

Our Response: This point about triplets has been raised by in **R#2 comment B** and we have tried to add the relevant discussion on this in as much details as possible, including a brief discussion of evidence from three independent experiments for the dominance of singlet-state dissociation, our own evidence from theory supporting this and a brief discussion of minor and major pathways that may include triplet states (where the latter clearly goes beyond the scope of our study, we believe). The XPS technique is very sensitive to the electronic structure of the involved complexes. The singlet pathway is not only ruled out by Wernet and Co-workers, but also by several other studies from Trushin and co-workers. We kindly refer the reviewer to **R#2 comment B** and our response therein.

Additionally, we made changes to the text and SI and added the discussion.

Regarding application of BS states, we would like to point out that BS DFT is applicable for ground state open shell species. We do not deal in our study with S_0 or as matter of fact even the S_1 state of $\text{Fe}(\text{CO})_4$ or $\text{Fe}(\text{CO})_5$. We have clearly omitted the discussion about dynamics of $\text{Fe}(\text{CO})_4$ fragment in S_1 and S_0 state and also from population analysis. Also, we are not sure about the sanctity of TDDFT on BS wavefunction or as a matter of fact any unrestricted wavefunction in such cases. We believe the error introduced cannot be estimated or controlled.

Explicit changes in the manuscript and the SI:

On page 2 of the manuscript, we have added a sentence referring also to the work of Ramasesha and co-workers [new ref.11].

On page 6 of the manuscript, we have added discussion of the triplet states and the intersystem crossing in this and other systems.

On page 8 in section I B of the SI, we have also added an extended discussion of the influence of the triplet states together with potential energy surfaces in new Figure S6

1c Single- vs. multideterminant description. Describing dissociation with DFT and TDDFT suffers from well-known problems due to the single determinant nature of the method. While the authors acknowledge this in the methods section and when describing the failed trajectories, the discussion does not go deep enough in my opinion. How can the authors be sure that the DFT/TDDFT electronic structures do not suffer from any problems before they fail? What criteria have they used to assess their validity? If any energetic or electronic structure uncertainties arise, how much would they affect the interpretation qualitatively and quantitatively? A detailed discussion will likely be too long for the main text, but would have its place in the SI of such a high level publication.

Perhaps the biggest methodological gap I view in this work is that since the TDDFT-based SHARC dynamics reveal a clear dissociation coordinate, why did the authors not follow up on this with wavefunction calculations along this coordinate? Even if the TDDFT had problems in the dissociation area, the wave function calculations in one dimension would provide a much clearer and complimentary picture of the dissociation process. To clarify this point, the authors could use wave function methods to check for multideterminantal character of the wavefunctions where prudent.

Our Response: This problem as pointed out by the reviewer is something that we are aware of, but the limitation of the theory available as we mentioned earlier rather confines us to the methodological choice. However, we have thoroughly checked the development of multi-reference character as the Fe-CO bond is cleaved. We performed a CASSCF(10,10)/NEVPT2 along that degree of freedom and plotted the energies in FigureS2. Additionally, here we present a measure of the multireference character, i.e. plot of the $(1-C_0^2)$ vs the Fe-CO bond distance. The C_0^2 is taken as the co-efficient of the closed shell CSF, $|2222200000\rangle$ contribution in the ground state wavefunction. As seen from plot presented below the development of multi-reference character in rather limited and hence our use of TDDFT is justified as far as single/multideterminant description is concerned.

However, as the scans are rigid and following the suggestions of the reviewer, we have conducted the same computation along one of the trajectories from the first burst.

We clearly see that though the $(1-C_0^2)$ remain nearly constant and below 0.2 mark most of the region, when Fe-C distance goes beyond 2.55 Å the system becomes multi-reference. This is also evident from the energies plot wrt to the distance where we see that S_1 and S_0 cross at a distance larger than 2.5 Å. Hence we are aware of this problem and we have now tried to improve the presentation in our manuscript. Based on this limitation we have excluded any data involving S_1 or S_0 dynamics, as based on this diagnostic we were already aware that any description of S_1 state is flawed as S_0 and S_1 energies become quasi-degenerate once the bond breaks. One serious limitations of our work is that we cannot describe the dynamics of the S_1/S_0 state of $\text{Fe}(\text{CO})_4$.

We have included a detailed discussion in SI along with these figures in the SI and hopefully highlight the limitation of our work, also mentioned in response to **R#1 comment 1**.

Explicit changes in the manuscript and the SI:

On page 5 of the manuscript, we have added a discussion on the above point.

On page 7 of the SI, we have added a discussion at the end of section IA and Figure S5 has been added and new caption reads "The plot $1-C^2_0$ vs Fe-C distance for taken from CASSCF(10e,10o) computation done for geometries from a excited state MD trajectory taken every 5 fs apart (left) and along a Fe-C scan(right) corresponding to Figure S3(b) "

R#3 comment 3: Scans

3a) In Figure S2, it is not clear how fine the scan is; the authors should revise these figures and include markers for the individual data points in the horizontal direction (e.g. on the PESs or as a separate horizontal line). The same applies to Figures S7 and S10.

Our response: We have done the scan with 0.05 Å. We have modified the figures of the scan accordingly by adding the points along the scan curves.

Explicit changes in the manuscript and the SI:

On page 5 and 19 of the SI, we have tried to improve the presentation of Figure S3 and Figure S15.

3b) Figure S2 also shows that the TDDFT- and NEVPT2-descriptions of the dissociative states differ. The authors should add a more detailed reasoning why they do not think that these differences matter. I would be more hesitant to treat the different crossing points of the dissociative states at (roughly) 1.85Å vs. 1.7Å and 2.15Å vs. 1.95Å as identical for the dissociation process.

Our Response: We agree that apart from similarities between TDDFT and NEVPT2, we clearly have difference. However, these differences are very difficult to validate in either way. Moreover, the states involved in the Frank-Condon region among which the differences arise are quasi-degenerate among themselves and hence they are often interchanged. These finer details are important but we do not feel that overall picture will change. We however plan to conduct full grid-based quantum dynamics with NEVPT2 potentials in the near future as a complimentary study to this work. However, at this point the effect of this is very hard to access.

Explicit changes in the manuscript and the SI:

On page 5 of the manuscript, we have added a section in line with our reply above.

R#3 comment 4: Dissociation criteria

The authors use as a dissociation criterion an Fe-C distance of 2.5 Å, while the PES scans show a significantly earlier crossing of the dissociative state(s) for the different methods. It is surprising that no electronic criterion is used, or even discussed why a purely distance-based choice was made. The authors should add a more detailed explanation here. How does the breathing mode with an amplitude of 0.3 Å and the equilibrium bond length of 1.8 Å fit in this picture – or phrased differently, what happens (electronically or structurally) in the species with bond lengths between 2.1-2.5 Å?

Our Response: We are grateful to the reviewer for bringing up this point, which was not clearly described in the original description. The value of 2.5 Å is selected by considerations of the running distance average in the individual trajectories, as displayed for a few cases in the present Figure S8. As stated in the caption of 2.5 Å is the lowest distance giving a unique time of dissociation for all trajectories. We have not been able to identify a useful electronic criterion in the trajectories. However, we see that from the PES scan and spot test that by 2.5 Å the dissociative MC states are well separated from the MLCT states and that the PES is almost flat, and yet we are within the single-reference regime.

We notice that in the sampling there are trajectories, where a 2.1-2.5 Å buffer region is required to avoid counting a motion which results in a reformation of the Fe(CO)₅ complex as a dissociation.

Explicit changes in the manuscript and the SI:

On page 7 of the manuscript, we have added a sentence on the 2.5 Å criterion

R#3 comment 5: Electronic structure discussion

The authors provide some electronic structure discussion, but there are several cursory statements like "As discussed earlier in detail from different points of view, the photodissociation happens by a transfer from non-dissociative to dissociative adiabatic states, which are associated with diabatic MLCT and MC states at long Fe-C distances." that do not do justice to the amount of data and level of analysis the authors have shown. I would encourage the authors to elaborate on the electronic structure aspects wherever they see fit, as they have done for explaining the minority pathway.

Our Response: We take into account the reviewer's suggestions and have elaborated the main text with electronic structure information. We have also now avoided references to earlier or later texts wherever possible.

Explicit changes in the manuscript and the SI:

On page 13 of the manuscript, we have removed the mentioned sentence and have reformulated it.

In SI we have introduced large sections which discuss the electronic structure in greater details and different level of theories.

R#3 comment 6: Discussion of changes in CO bonding I may have overlooked this, but I did not see any discussion of changes in the CO bonding situation. This would be very valuable to experimental chemists since the CO stretching frequency is a sensitive probe of electronic structure and geometry, and can be monitored with high accuracy. Can the authors extract or extrapolate which changes they expect in the vibrational spectra along the dissociation process?

Our Response: Yes, we had not looked into this aspect. We agree that this is the very valuable point that has experimental implication.

In New Table S4, we see that the frequencies corresponding to CO vibration are systematically blue shifted in the MLCT excited state at for the vertical geometry and the S_5 optimized geometry. This can be thought of as the effect of excitation of an electron from an Fe d orbital to the CO π^* MOs. Thus, the system residing in a bound MLCT state could be investigated using ultra-fast time resolved IR experiments.

Explicit changes in the manuscript and the SI:

On page 13 of the manuscript, we have added a sentence on the Fe-C and CO vibrations, and the possibility of detecting electronic changes.

On page 20 of the SI, we have added a new Table S4 containing a normal mode analysis of $\text{Fe}(\text{CO})_5$ in the S_0 and S_5 states, which also contains the CO stretching frequencies.

REVIEWERS' COMMENTS

Reviewer #1 (Remarks to the Author):

The authors have made several important additions to the manuscript to address my comments, as well as those of reviewers 2 and 3. I believe it is now in good shape for publication in Nature Communications.

Reviewer #2 (Remarks to the Author):

I thank Banerjee et al for addressing all the comments from my previous review. The changes in the manuscript and especially the additional results and discussion included in the supporting information have improved the already substantial quality of the present work. I think the manuscript will be interesting for the readership of Nature Communications and should thus be accepted for publication.

A few minor points that the authors should nevertheless still address are as follows.

p. 5, l. 134: It should be clarified whether the cuts in Figure S3 in the SI are from unrelaxed or relaxed scans. The same applies to the discussion of the other potential energy curves in this work.

p. 16, l. 463: it is stated that vibrational modes at the B3LYP/ccpVDZ level were obtained based on a CASPT2/TZVP optimized geometry. I assume the modes were only calculated after reoptimization at the DFT level, which should be mentioned.

p. 27, Figure 3: Have the colors between panels b and c accidentally been reversed? In b, the dissociative population is red and increases over time, while in c, the red curves decrease over time (blue curves vice versa).

SI, p2: The different behavior of NEVPT2 and "TDDFT" is discussed for the studied system. For clarity, it should be added that the TDDFT behavior is only derived from the three functionals mentioned in Table S1, and may not be a general behavior of TDDFT.

SI, p2, second paragraph: It is stated that “Fe(CO)₄ has a triplet ground state and thus with such systems the singlet ground state often has a multideterminantal wavefunction”. I assume the “singlet ground state” then refers to the lowest excited state of singlet spin multiplicity. Usually, the term ground state is reserved to the state of overall lowest energy, so maybe the wording should be changed.

SI, p 8/9: Figure S6 includes TDDFT potential energy curves, that, however, do not seem to be mentioned in the accompanying discussion in Section B on p. 8. Perhaps the authors could add a few sentences comparing the TDDFT and NEVPT2 curves in light of their discussion on excluding the triplet states from the simulation.

Reviewer #3 (Remarks to the Author):

The authors have addressed all major points of criticism in the revised version of the manuscript. I thoroughly appreciated the detail given in the responses and added in the main text and SI. The authors have added extensive discussions regarding methodological aspects and comparison with experimental insights. Furthermore, results from additional calculations are now shown and discussed in the Supporting Information. In my view, the presentation of the results is now significantly improved and contextualised better.

With the revision, the authors have lifted an already good paper on a highly challenging topic to a much higher level that will likely be used as a reference point for other theoreticians in the field. I recommend publication as is.

Reviewer #2:

I thank Banerjee et al for addressing all the comments from my previous review. The changes in the manuscript and especially the additional results and discussion included in the supporting information have improved the already substantial quality of the present work. I think the manuscript will be interesting for the readership of Nature Communications and should thus be accepted for publication.

Our Response: We thank the reviewer for recommending the manuscript for publication and for additional valuable remarks to improve the manuscript.

A few minor points that the authors should nevertheless still address are as follows.

p. 5, l. 134: It should be clarified whether the cuts in Figure S3 in the SI are from unrelaxed or relaxed scans. The same applies to the discussion of the other potential energy curves in this work.

Our Response: The scans are unrelaxed scans and these are mentioned in page 8, 9 and 13. We have also modified the text to further clarify this.

Explicit changes in the manuscript and the SI:

We have additionally added a sentence in the Methods section in page 17.

The potential energy cuts as shown in Supplementary Figures 3, 6, 10, 11, and 15 are produced by unrelaxed (rigid) scan of one or two C-O bonds, respectively.

p. 16, l. 463: it is stated that vibrational modes at the B3LYP/ccpVDZ level were obtained based on a CASPT2/TZVP optimized geometry. I assume the modes were only calculated after re-optimization at the DFT level, which should be mentioned.

Our Response: We have used the CASPT2/TZVP geometry and performed frequency calculations with B3LYP/ccPVDZ. The CASPT2(12,12)/TZVP geometry is used as it is a gold standard in reproducing the geometry of FeCO₅. Though the frequency and optimization are done at different levels of theory no imaginary frequency was found, and performing frequency computation on B3LYP/ccPVDZ optimized geometry, which differs in the bond lengths by less than 0.01 Å from CASPT2/TZVP and gives essentially the same normal modes and vibrational spectra, differing only by ~20 cm⁻¹ in the five Fe-CO stretching modes, which play the most important role in the dynamics,

Explicit changes in the manuscript and the SI:

We have added a discussion on Methods section on page 15.

Though the frequency calculation and optimization are done at different levels of theory no imaginary frequency was found, and performing frequency computation on the B3LYP/cc-PVDZ optimized geometry gives essentially the same vibrational spectra, differing only by ~20 cm⁻¹ in the five Fe-CO stretching modes, which play the most important role in the dynamics.

p. 27, Figure 3: Have the colors between panels b and c accidentally been reversed? In b, the dissociative population is red and increases over time, while in c, the red curves decrease over time (blue curves vice versa).

Our Response: We are extremely sorry for the mistake. We thank the reviewer for rightly pointing out this swapping of color.

Explicit changes in the manuscript and the SI: Figure 4(old Figure 3) has been changed with the right colors.

SI, p2: The different behavior of NEVPT2 and “TDDFT” is discussed for the studied system. For clarity, it should be added that the TDDFT behavior is only derived from the three functionals mentioned in Table S1, and may not be a general behavior of TDDFT.

Our Response: We agree that the behavior is limited to the three functionals and we have made the necessary changes in the SI

Explicit changes in the manuscript and the SI:

In SI in page 2 section I we have added a note on this.

We would also like to mention here that the variety of functionals explored for TDDFT is rather limited in the present study, and the aforementioned trend is limited to the three functional mentioned in Supplementary Table 1. This may change for different functionals.

SI, p2, second paragraph: It is stated that “Fe(CO)₄ has a triplet ground state and thus with such systems the singlet ground state often has a multideterminantal wavefunction”. I assume the “singlet ground state” then refers to the lowest excited state of singlet spin multiplicity. Usually, the term ground state is reserved to the state of overall lowest energy, so maybe the wording should be changed.

Our Response: We have changed the wording from single ground state to singlet S₀ state.

Explicit changes in the manuscript and the SI:

In SI page 2 we have made this change.

SI, p 8/9: Figure S6 includes TDDFT potential energy curves, that, however, do not seem to be mentioned in the accompanying discussion in Section B on p. 8. Perhaps the authors could add a few sentences comparing the TDDFT and NEVPT2 curves in light of their discussion on excluding the triplet states from the simulation.

Our Response: This TDDFT scan with triplets are essentially similar to the NEVPT2 plot and the crossing between ¹MC state and ³MLCT happens in a region where dissociation has already been initiated and has a dissociative slope.

Explicit changes in the manuscript and the SI:

In SI at the end of page 8, we have added a discussion on this.

The scan at the TDDFT/CAM-B3LYP level in Supplementary Figure 6 also gives the same overall behavior for the singlet and triplet manifolds although the 3MLCT states are less dispersed than the 1MLCT states. Notice also that the crossings between the 1MC and 3MLCT states occur in a region where the dissociation has already been initiated.